# Quantifying DNA replication speeds in single cells by scEdU-seq

Jeroen van den Berg [1,5,6] ✉, Vincent van Batenburg [1,5], Christoph Geisenberger [1,2], Rinskje B. Tjeerdsma [3], Anchel de Jaime-Soguero [4], Sergio P. Acebrón [4], Marcel A. T. M. van Vugt[3] & Alexander van Oudenaarden [1,6] ✉

In a human cell, thousands of replication forks simultaneously coordinate duplication of the entire genome. The rate at which this process occurs might depend on the epigenetic state of the genome and vary between, or even within, cell types. To accurately measure DNA replication speeds, we developed single-cell 5-ethynyl-2′-deoxyuridine sequencing to detect nascent replicated DNA. We observed that the DNA replication speed is not constant but increases during S phase of the cell cycle. Using genetic and pharmacological perturbations we were able to alter this acceleration of replication and conclude that DNA damage inflicted by the process of transcription limits the speed of replication during early S phase. In late S phase, during which less-transcribed regions replicate, replication accelerates and approaches its maximum speed.

Before cell division, the genome has to be faithfully replicated and DNA replication errors have to be averted to prevent developmental defects and tumorigenesis[1–3]. Genetics and biochemistry have revealed many DNA replication factors and their cooperation to ensure high-fidelity duplication of genomes. To probe DNA replicative processes in a genome-wide manner, DNA sequencing methods have been used to unravel DNA replication dynamics such as replication timing, replication fork directionality, origin and DNA polymerase usage[4–7]. Furthermore, single-molecule approaches have been widely used to monitor the behavior of individual replication forks, including replication speed, as well as fork stalling through detection by microscopy[8–10] or long-read sequencing[11–13]. However, these methods randomly sample molecules from a large population of cells and are therefore insensitive to heterogeneity in replication dynamics between individual cells. We describe a method, single-cell 5-ethynyl-2′-deoxyuridine sequencing (scEdU-seq), that enables a high-resolution single-cell investigation of DNA replication fork dynamics.

## Results

### scEdU-seq reveals DNA replication profiles through S phase

To measure the heterogeneity of DNA replication fork dynamics, we developed scEdU-seq, a sequencing method to identify replicated nascent DNA in individual cells. scEdU-seq relies on metabolic labeling with the nucleotide analog 5-ethynyl-2′-deoxyuridine (EdU) and affinity capture of newly synthesized DNA fragments (Fig. 1a). We use copper(I)-catalyzed azide-alkyne cycloaddition click chemistry to covalently link a biotin moiety to the uracil base[14]. Following click, we sort single cells into a 384-well plate for single-cell processing. Subsequently, we digest the single-cell genome using a restriction enzyme (NlaIII) and end-repair fragments (large Klenow fragment and polynucleotide kinase), which are ligated to a T7 promoter containing adapters, cell-specific barcodes and a unique molecular identifier (UMI)[15]. After pooling cells, we biotin-capture the EdU-containing DNA molecules and release the non-EdU-modified strand by heat denaturation. Next, we regenerate the complementary strand via

[1]Oncode Institute, Hubrecht Institute-KNAW (Royal Netherlands Academy of Arts and Sciences) and University Medical Center Utrecht, Utrecht, The Netherlands. [2]Pathologisches Institut, Ludwig-Maximilians-Universität, Munich, Germany. [3]Department of Medical Oncology, University of Groningen, University Medical Center Groningen, Groningen, The Netherlands. [4]Centre for Organismal Studies (COS), Heidelberg University, Heidelberg, Germany. [5]These authors contributed equally: Jeroen van den Berg, Vincent van Batenburg. [6]These authors jointly supervised this work: Jeroen van den Berg, Alexander van Oudenaarden. ✉e-mail: j.berg@hubrecht.eu; a.vanoudenaarden@hubrecht.eu

Klenow-mediated primer extension, followed by linear amplification via T7-dependent transcription. Finally, linearly amplified RNA is converted to complementary DNA by reverse transcription (RT) and amplified using a polymerase chain reaction (PCR) to prepare for Illumina sequencing.

We compare scEdU-seq with Repli-Seq[7] in human RPE-1 cells expressing human telomerase reverse transcriptase (hTERT) and find substantial overlap of the DNA replication profiles (Fig. 1b and Extended Data Fig. 1a–d). A feature of scEdU-seq is profiling non-nascent DNA in conjunction with nascent DNA (Extended Data Fig. 1e,f). As expected, we observe an anticorrelation between nascent and non-nascent DNA from the same sample. Next, we set out to generate DNA replication profiles in single cells, using fluorescence-activated cell sorting (FACS) enrichment for S-phase cells (based on 4′,6-diamidino-2-phenylindole (DAPI)) labeled with 15 min of EdU (Extended Data Fig. 1g). We observe that our sorting gates include all S-phase cells, because we detect no scEdU-seq⁺ cells at the extremities of the DAPI gates (Extended Data Fig. 1h,i). We reconstruct progression through S phase by ordering cells based on the overlapping scEdU-seq signal between single cells. Ordering of single cells by S-phase progression relies on the assumption that cells adhere to a similar replication timing, which has previously been shown for RPE-1 cells[16]. We sought to find an overall S-phase progression position for all sampled cells. First, we compare cells in a pairwise manner using the overlap coefficient, giving us a pairwise similarity score (Extended Data Fig. 1j and Methods). These pairwise scores are converted to distances and reduced using a one-dimensional uniform manifold approximation and projection (UMAP) (Fig. 1c). The average bootstrapped order of the cells on this UMAP line is the S-phase progression position. To validate the inferred S-phase progression, we perform scEdU-seq in cells expressing the fluorescent ubiquitination-based cell cycle indicator (FUCCI) reporter system, a fluorescent read-out reflecting cell-cycle stage, allowing us to record the S-phase position of each sequenced single cell. We observe that scEdU-seq based S-phase progression accurately reflects the position in the cell cycle shown by FUCCI[17] reporters and DNA content (Fig. 1d,e and Extended Data Fig. 1k). This shows that we can use scEdU-seq based S-phase progression to order cells throughout DNA replication. Implementing single-cell S-phase progression ordering, we construct DNA replication tracks over the entirety of S phase from an ensemble of 1,343 RPE-1 hTERT FUCCI cells (Fig. 1f; genome-wide visualization provided at https://sceduseq.eu/). scEdU-seq based S-phase ordering is consistent with published bulk DNA replication timing using cell-cycle sorted populations (Pearson $\rho$ = 0.8, Extended Data Fig. 1l,m)[18]. In addition, we observe that the start of scEdU-seq tracks (early S-phase progression) overlaps with genome positions that were identified by 5-ethynyl-2′-deoxyuridine sequencing with hydroxyurea (EdUseq-HU) as initiation zones of DNA replication in RPE-1 cells (Fig. 1f)[19]. As expected, we observe the greatest overlap between EdUseq-HU and scEdU-seq in the initial 5% of S-phase progression (Pearson $\rho$ > 0.65; Extended Data Fig. 1n).

Regulation of the number of DNA replication forks is essential to limit replication stress and genomic instability[19–21]. Experimental quantification of the number of DNA replication forks is challenging. Initial efforts with imaging techniques[8] resulted in an estimation of 5,000 DNA replication forks per human cell[22]. Quantification of individual DNA replication forks from single cells requires segmentation of the scEdU-seq signal into blocks of DNA replication tracks. To achieve these DNA replication fork calls, we fit a hidden Markov model (HMM) per cell to identify stretches that were generated by the same replisome (Methods and Extended Data Fig. 2a–f). We observe that RPE-1 cells have ~4,500 forks per cell (Fig. 1g), consistent with imaging studies[22]. Moreover, the number of DNA replication forks per chromosome correlates with the length of each human chromosome (Extended Data Fig. 2g). As expected, chromosomes 10 and 12 display an elevated number of DNA replication forks because these chromosomes are (partially) amplified in RPE-1 cells (chr10q and full chr12). The DNA replication fork number between single chromosomes follows the same trend over S phase suggesting similar regulation of DNA replication per chromosome (Extended Data Fig. 2h).

In addition to quantification of the number of forks, we use these analyses to compute the sensitivity of scEdU-seq. To assess sensitivity, we quantify the number of detected replication forks per cell following downsampling of unique reads (Fig. 1h and Extended Data Fig. 2i). We find that downsampling reads from cells results in a decrease in the number of detected forks; however, this decrease only becomes apparent after removing 25–30% of unique reads from single cells. We observe that for the majority of cells, we still detect the vast majority of DNA replication forks after removing 10,000 unique reads (Extended Data Fig. 2i), which is in line with the number of detected reads per replication fork (Extended Data Fig. 2f). To further explore scEdU-seq sensitivity, we attempt to compare the ability of scEdU-seq and single-cell Repli-Seq (scRepli-Seq) to detect DNA replication forks. Using a similar analysis, we observe twice as many DNA replication forks genome-wide in scEdU-seq in similarly staged cells during S phase (Extended Data Fig. 3a,b). For a representative genomic locus, we find that certain regions contain DNA replication forks in scEdU-seq that are not or less frequently detected in scRepli-Seq (Extended Data Fig. 3c,d). Taking these results together, scEdU-seq allows high-sensitivity profiling of DNA replication forks in single cells.

## Double-pulse scEdU-seq allows DNA replication speed estimate

We have shown that scEdU-seq is able to detect the majority of DNA replication forks with high resolution. Next, we set out to determine replication speeds in single cells using scEdU-seq. Although replication speeds can be estimated from replication track widths, these depend on the sequencing depth (Extended Data Fig. 4a). By contrast, a double-pulse EdU-labeling strategy leads to increased accuracy of replication speeds in single cells and is less sensitive to the unique reads recovered per cell (Extended Data Fig. 4a–c).

**Fig. 1 | scEdU-seq reveals ordered DNA replication profiles throughout S phase. a**, Representation of the scEdU-seq protocol. **b**, $Z$-scored genome-coverage tracks of log$_2$(fold change) (early/late) S-phase samples for both scEdU-seq (500 cell bulk, upper) and Repli-Seq (400,000 cell bulk, lower) treated with 120 min of EdU. **c**, Dimensional distance between single cells by UMAP. Each dot is a single cell, lines indicate nearest neighbors, dots are colored by S-phase progression, DNA content (DAPI) or FUCCI markers. **d**, Scatter plot showing FUCCI reporters pseudo-colored by S-phase progression, determined using scEdU-seq tracks from a 15-min EdU pulse of cycling RPE-1 cells overlaid on the cell-cycle distribution of control RPE-1 cells (gray). **e**, Rolling mean of the $z$-scored fluorescence intensity of FUCCI reporters and DNA content (DAPI, $y$ axis) versus S-phase progression ($x$ axis) based on scEdU-seq tracks (single EdU pulse (15 min)). The ribbon indicates the standard deviation.

**f**, Heatmap of scEdU-seq from single EdU pulse (15 min) maximum normalized log counts for 1,343 RPE-1 hTERT FUCCI cells ordered according to S-phase progression ($y$ axis) and binned per 400 kb bins ($x$ axis) for a 50 megabase region of chromosome 2. Heatmap showing log$_2$(fold ratio) of early to late Repli-Seq indicating replication timing (upper) and a bar graph showing the replication origins (EdUseqHU[19]) of the same stretch of chromosome 2 (lower). Scaled $z$-scored intensities of FUCCI reporters and DNA content (DAPI) ordered by S-phase progression are shown on the right. **g**, Number of forks ($y$ axis) per cell versus S-phase progression ($x$ axis). The line represents the rolling-window median and the ribbon indicates the 95% confidence interval of the windows. **h**, Subsampling of unique reads per cell ($x$ axis) versus detected number of DNA replication forks per single cell ($y$ axis) for 15-min EdU-treated RPE-1 hTERT cells.

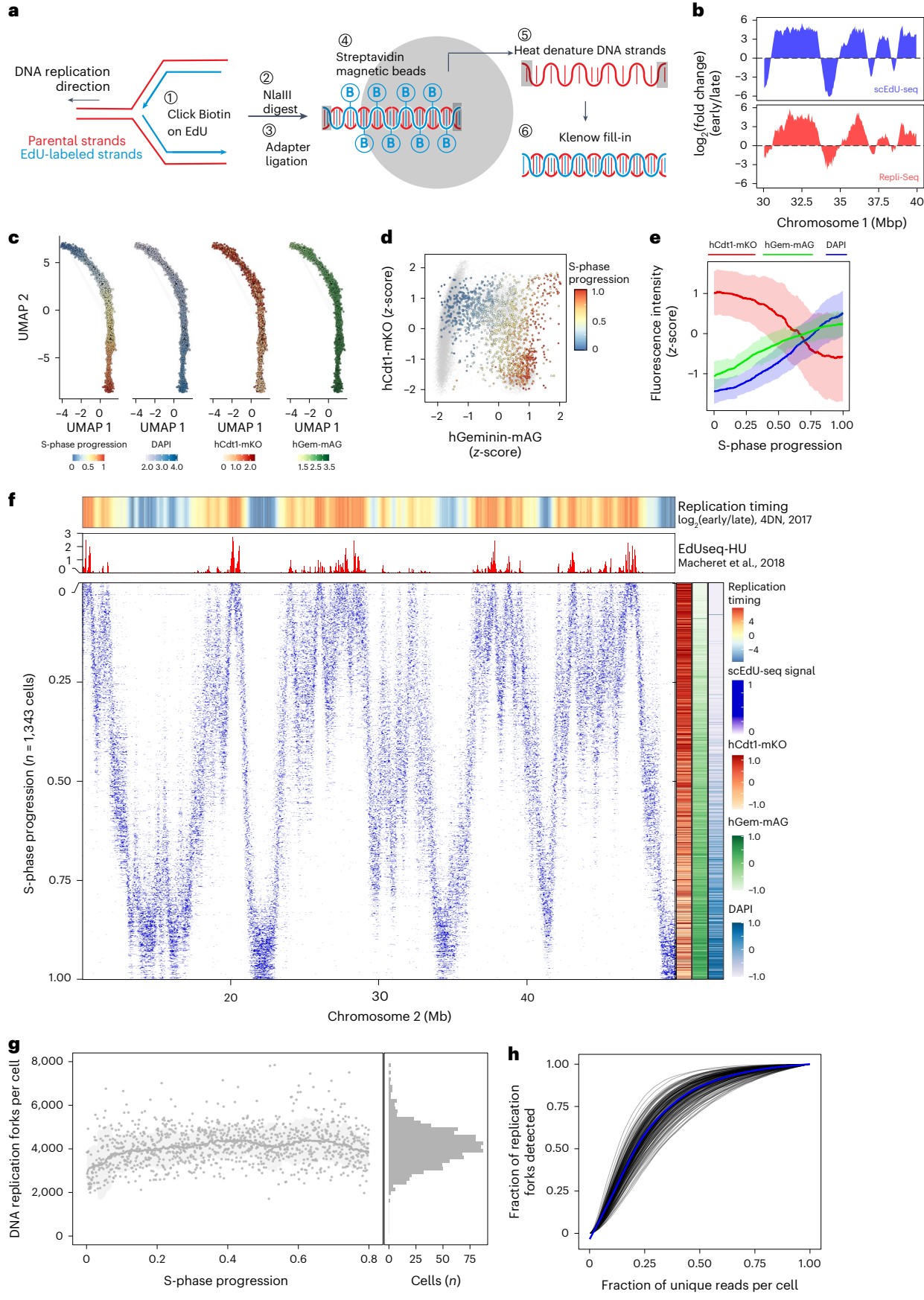

After receiving two EdU pulses, the genome of a single cell in S phase is decorated with patches of EdU that are separated by distance $\Delta x$. The average replication speed is approximated by dividing this distance, $\Delta x$, by the EdU pulse center-to-center timespan $\Delta t$ (labeling times). To systematically analyze these data, we use the pair correlation function[23], which is defined as the distribution of all pairwise distances between EdU-containing reads in a single cell (Fig. 2a, right). For two EdU pulses, we expect this distribution to contain three main features that reflect different structures in the data. First, pairs of reads that were labeled within one pulse would generate a distribution of short distances (Fig. 2a, intrapulse distances in yellow). Next, read pairs from the same replication fork labeled in separate pulses would generate a distribution around distances at $\Delta x = \bar{\Delta} x$ (Fig. 2a, interpulse distances in cyan). Finally, pairs of reads labeled in either pulse from different replication forks would yield a uniform background (Fig. 2a, interfork distances in pink). From the location of the interpulse distance distribution, we can estimate the average DNA replication fork progression in a single cell. We compute the average DNA replication fork speed for each cell by dividing the average distance traveled $\bar{\Delta} x$ by the time between pulses $\Delta t$. Experimental pair correlations consistently behaved as expected. A single cell exposed to a single EdU pulse shows one maximum at $\Delta x = 0$ (Fig. 2b and Extended Data Fig. 4d,e). Conversely, cells exposed to a double EdU pulse ($\Delta t = 45$, 75 or 105 min) also show a second maximum. As expected, this second maximum shifts to larger values of $\Delta x$ as $\Delta t$ is increased (Fig. 2b, upper and Extended Data Fig. 4f).

We find that using single cells is critical; adding together pairwise distances from different cells drastically alters the signal in the pair correlation, creates noise and hinders the speed measurement (Fig. 2c; 1 cell versus 100 cells). This occurs even at low cell numbers (5–100 cells; Extended Data Fig. 4g). In addition, downsampling unique reads from a single cell does not hamper detection of the second maxima in the pair correlation (Extended Data Fig. 4g). This implies that many DNA replication forks contribute to the second maxima. A fraction of the pairwise distances between read pairs originates from two different forks, which potentially confounds our analyses (Fig. 2a; distances indicated in pink). Conceptually, we expect these effects to result in a background signal of the pair correlation function. To confirm this, we use our single-pulse data and recover a near-uniform distance distribution between forks per cell (Fig. 2d). This shows that a signal from separate forks does not interfere with detection of the second maximum ($\bar{\Delta} x$). In line with this observation, we do not detect a second maximum in the pair correlation function for a single pulse (Extended Data Fig. 4h,i).

To quantify DNA replication speeds from these data, we analyze the position of the second maximum ($\bar{\Delta} x$) of single cells exposed to a double EdU pulse (Methods and Fig. 2e, upper). As expected, this second maximum increases $\bar{\Delta} x$ as a function of $\Delta t$ ($\Delta t = 45$, 75 or 105 min) and appears to increase linearly along $\Delta x$ with the increase in $\Delta t$ (Fig. 2e and Extended Data Fig. 4f). Indeed, when $\bar{\Delta} x$ is divided by $\Delta t$ we find similar DNA replication speeds for all labeling strategies (Fig. 2e, lower). This shows that the increase in $\bar{\Delta} x$ as a function of $\Delta t$ is caused by progressing DNA replication forks. Pairwise distances per chromosome show similar distributions of $\bar{\Delta} x$ compared with all chromosomes combined (Fig. 2f).

The overall distribution of distances between reads is the result of multiple sources (Fig. 2a). To quantitatively model the individual sources, we use a mixture model. This allows estimation of the parameters of individual sources without the requirement for a priori assignment of each distance to one of the sources. We use a normal distribution to fit the interpulse distance (speed) component. Subsequently, we can obtain corresponding confidence intervals by bootstrapping with an expectation maximization algorithm and fit the model (Extended Data Fig. 5a–c)[24]. Finally, we observe a slight bias

in DNA replication speeds as the timespan between labeling pulses increases (Fig. 2e,f). Simulations of double-pulse data show that higher speeds result in longer replication tracks, which have a greater weight and therefore contribute more to the resulting pair correlation (Extended Data Fig. 5d–g). We correct for this effect using the simulated data (Extended Data Fig. 5h). Taken together, we can use double-pulse EdU labeling combined with pair correlation analysis to identify DNA replication speeds in single cells.

## Transcription limits DNA replication speeds in early S phase

We can measure DNA replication speeds in single cells by using a double EdU pulse in combination with a mixture model. Representative cells labeled with a double EdU pulse ($\Delta t = 75$ min) demonstrate that the mixture model (Fig. 3a, red line) accurately describes the experimental pair correlation (black line). Overall, we find DNA replication speeds in the expected range described in literature[25]. Unexpectedly, we observe great variability in replication speeds between individual cells (~1.5-fold difference, Fig. 2c,d). A large part of this variability is explained by the position of single cells in S phase. We observe a steady increase in DNA replication speeds suggesting acceleration of replication throughout S phase (Fig. 3b).

We can also estimate replication speeds with single-pulse labeling by quantifying the width of DNA replication tracks (Methods). We find similar DNA replication speeds and acceleration of replication throughout S phase in single-pulse data (Methods and Extended Data Fig. 6a–c). In addition, we confirm these phenotypes by DNA fiber analysis on sorted early and late S-phase RPE-1 hTERT cells (Fig. 3c,d and Extended Data Fig. 6d). Finally, we profile human-induced pluripotent stem cells and observe a similar increase in replication speeds during S phase (Extended Data Fig. 6e–g). Moreover, a recent study using long-read sequencing describes similar DNA replication speeds as well as an increase in DNA replication speeds throughout S phase[26]. These independent observations support the validity of the double-pulse EdU experiments.

Variability in DNA replication rates has been observed since the 1970s (refs. [27–29]). In human cells, the acceleration of replication speeds throughout S phase has not been previously observed and the mechanism behind the reduced DNA replication speeds in early S phase remains elusive. Previous studies have shown that early replicating DNA is close to actively transcribed regions[30]. Because we observed the lowest replication speeds in early S phase, we hypothesize that lower speeds might be caused by transcription. To quantify transcription levels across S phase, we use single-cell nascent RNA sequencing data on RPE-1 cells (Extended Data Fig. 7a–d)[31]. As expected, we observe the highest levels of transcription in regions of the genome that overlap with the scEdU-seq signal at the start of S phase (Fig. 3e,f). In addition, both the number of transcribed regions and transcription levels decrease as DNA replication progresses over S phase. The presence of high levels of transcription in early S phase correlates with lower DNA replication speeds (Fig. 3e,f).

Indeed, transcribed regions in early S phase are replicated even more slowly than nontranscribed regions (Fig. 3e), which implies that transcription limits DNA replication speeds in early S phase. Nonetheless, we still observe the acceleration of DNA replication speeds over the course of S-phase progression outside transcribed regions (Extended Data Fig. 7e). This suggests that other factors control DNA replication speeds. To address this, we profile RPE-1 cells using chromatin immunocleavage sequencing[15] for H3K36me3, H3K27me3 and H3K9me3 modifications, which align with the expected patterns of DNA replication timing[32] (Extended Data Fig. 7f,g). We observe that both H3K36me3 and transcribed regions display lower DNA replication speeds (Fig. 3g), confirming our initial observation. Conversely, we find that H3K9me3-repressed chromatin confers higher DNA replication speeds. In addition, we observe that within H3K27me3 chromatin, DNA replication speeds do not drastically differ.

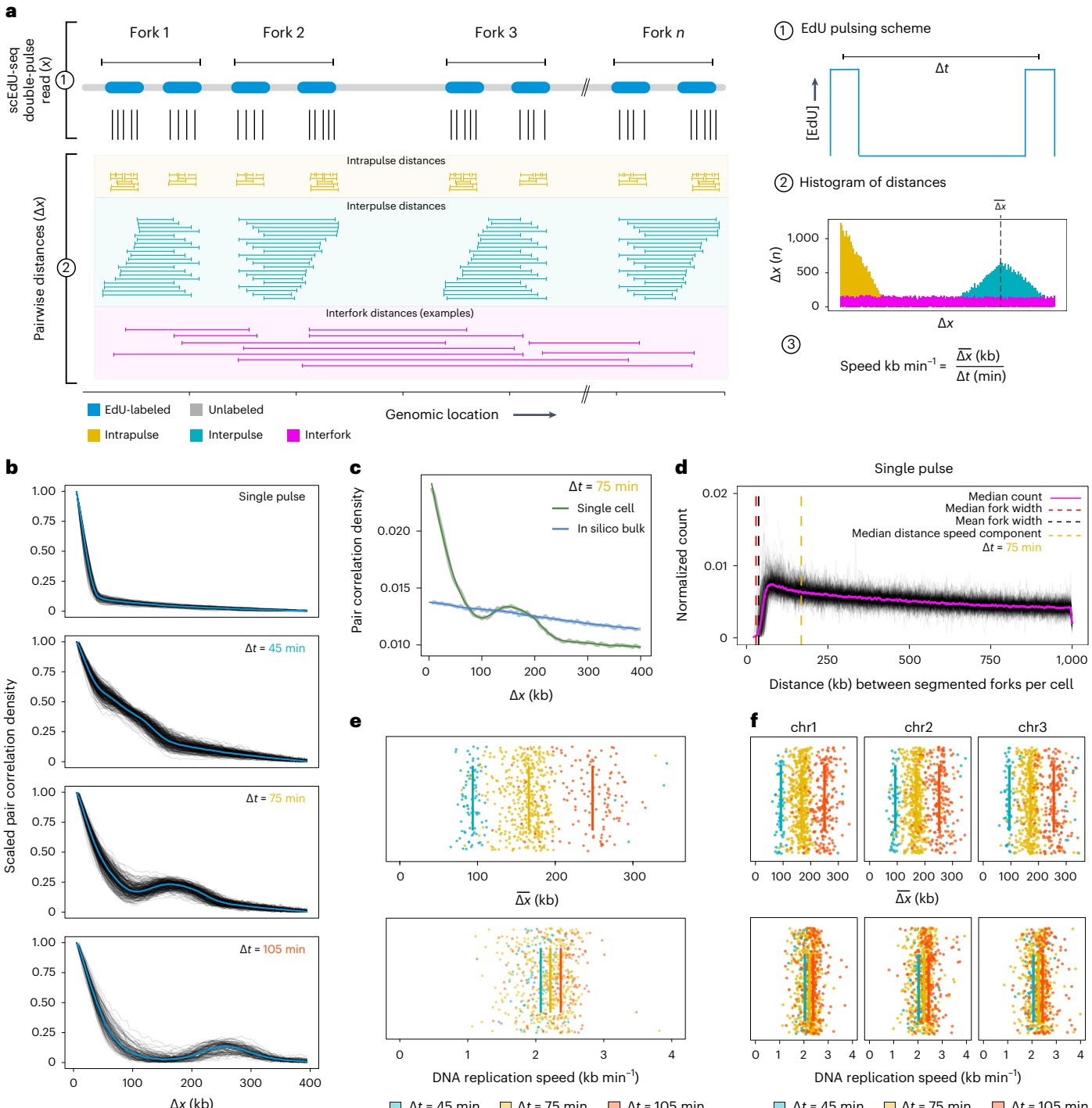

**Fig. 2 | Double-pulse scEdU-seq allows DNA replication speed assessment.** **a**, Schematic representation of the double-pulse labeling scheme and subsequent analysis. **b**, Line plots of the pair correlations of the single-pulse and double-pulse labeling scheme with $\Delta t$ = 45 min ($n$ = 376), $\Delta t$ = 75 min ($n$ = 347) and $\Delta t$ = 105 min ($n$ = 149). Each line is a single RPE-1 cell, where the $x$ axis shows the binned distance $\Delta x$ (kb) and the $y$ axis is the range-scaled density. A blue line indicates the mean density per bin. **c**, Pairwise distance between all reads from a $\Delta t$ = 75 min scEdU-seq experiment of a single cell (green) and in silico bulk (100 cell bulk, blue). The thick line represents raw data and the thin line represents a LOESS regression of the raw data. **d**, Normalized count ($y$ axis) of binned pairwise distances ($x$ axis) between HMM-segmented pulses. Each line is a single cell and the magenta line indicates the median normalized count.

Black and red lines indicate the median and mean size of a segmented fork in a 15 min EdU single-pulse scEdU-seq experiment. The yellow line indicates the median distance for the speed component in $\Delta t$ = 75 min scEdU-seq data. **e**, DNA replication distance estimates ($\bar{\Delta}x$) in kb for cells ($\Delta t$ = 45 min ($n$ = 57), $\Delta t$ = 75 min ($n$ = 299) and $\Delta t$ = 105 min ($n$ = 103)) from **b** colored by labeling scheme. Upper, Ticks indicate the averages per labeling scheme. Lower, Distance estimates corrected for labeling resulting in DNA replication speeds (kb min$^{-1}$). **f**, DNA replication speed estimates ($\bar{\Delta}x$) per chromosome ($\Delta t$ = 45 min ($n$ = 57), $\Delta t$ = 75 min ($n$ = 299) and $\Delta t$ = 105 min ($n$ = 103)) from **b** colored by labeling scheme. Upper, Ticks indicate the average per labeling scheme. Lower, Distance estimates corrected for labeling scheme resulting in DNA replication speeds (kb per min per chromosome).

If active RNA polymerase II (RNAPII) transcription reduces DNA replication speeds, we would expect an increase in DNA replication speeds in early S phase by inhibiting transcription. To assess DNA replication speeds without active RNAPII transcription, we treat cells with the transcription inhibitor 5,6-dichlorobenzimidazole-1-β-D-ribofuranoside (DRB, 60 min) (Extended Data Fig. 8a–e) between the two EdU pulses, which does not alter either initiation zones or replication timing (Extended Data Fig. 8f,g). We observe an overall increase in DNA replication speeds in RNAPII-inhibited cells versus dimethylsulfoxide (DMSO)-treated cells (Fig. 3h, bottom). Furthermore, this difference in speed results from increased DNA replication speeds during early S phase (Fig. 3h, upper). Finally, we analyze DNA replication speeds in DRB-treated RPE-1 cells for transcribed regions as well as H3K36me3 chromatin (Fig. 3i). We find that we can increase DNA replication speeds in transcribed as well as H3K36me3 chromatin using the RNAPII inhibitor DRB. We find that the RNAPII inhibitor almost completely removes the influence of transcription and H3K36me3 on DNA replication speeds. In summary, DNA replication accelerates over S phase, in part as a result of RNAPII transcription decreasing replication speeds during early S phase.

## Transcription-coupled damage decreases DNA replication speed

RNAPII activity has been correlated with a variety of types of DNA damage; for example, by generating single-strand breaks through topoisomerase I cleavage complexes or repair of bulky adducts by transcription-coupled nucleotide excision repair[33,34]. Moreover, conflicts between the DNA replication fork and transcription machinery lead to the formation of RNA:DNA hybrids, which result in double-strand breaks if improperly handled[35]. Indeed, short inhibition of RNAPII (1 h) during S phase results in a reduction in DNA damage as assayed by flow cytometry (Fig. 4a; γH2AX). This suggests that RNAPII activity, at least in part, causes transcription-coupled DNA damage during S phase.

The activity of the DNA damage sensor poly(ADP-ribose) polymerase 1 (PARP-1) is stimulated by a wide variety of DNA damage lesions[36]. Because PARP-1 activity has previously been linked to DNA replication speeds[25], we reason that the decrease in DNA replication speed in early S phase might be caused by transcription-coupled DNA damage and subsequent PARP activation. We observe a decrease in the level of pan ADP-ribose, the modification deposited by PARP enzymes, upon PARP inhibition (Fig. 4a, pan ADP-ribose). This suggests that RNAPII transcription during S phase not only induces DNA damage, but also activates PARP.

To explore how transcription-coupled DNA damage might affect DNA replication speed in single cells, we make use of the PARP inhibitor Olaparib. First, we treat wild-type (WT) RPE-1 cells with a PARP inhibitor (PARPi) and observe very similar DNA replication speed behavior compared with RNAPII inhibition (Fig. 4b and Extended Data Fig. 9a–e).

Overall DNA replication speeds are higher in PARP-inhibited cells without altering either initiation zones or replication timing (Extended Data Fig. 9f). In addition, the most notable difference in DNA replication speeds occurs in early S phase suggesting a connection to RNAPII transcription. To validate these findings, we use the single-pulse strategy and fork-width analysis and find higher DNA replication speeds, specifically in early S phase, in PARPi treatment compared with DMSO treatment (Extended Data Fig. 9g,h).

To further address the role of PARP activity in regulating DNA replication speeds, we hyperactivate PARP-1 by generating an RPE-1 cell line in which the gene *XRCC1* was knocked out (XRCC1Δ RPE-1) (Extended Data Fig. 9i,j). XRCC1 protein is required for efficient repair of DNA damage. In the absence of this protein, an increase in steady-state levels of DNA damage causes PARP hyperactivation, which eventually leads to cerebral ataxia[37] (Fig. 4c–e). XRCC1Δ cells have a lower proportion of EdU+ cells compared with WT cells, which is partially mitigated by PARPi treatment (Fig. 4f). This implies that excessive PARP signaling in XRCC1Δ cells result in lower DNA replication speeds. In line with this observation, XRCC1Δ cells display overall lower DNA replication speeds compared with WT cells (Fig. 4g and Extended Data Fig. 9k,l). In contrast to RNAPII or PARP inhibition, we observe a ubiquitous decrease in DNA replication speeds in all cells, not just early S-phase cells. This suggests that hyperactivation of PARP, outside transcribed regions, results in lower DNA replication speeds. In addition, we can rescue the global decrease in DNA replication speeds in XRCC1Δ RPE-1 by PARPi treatment (Fig. 4h).

In addition to these changes in DNA replication speeds, we also observe that the variability in speed is altered within an individual cell. We find that inhibiting transcription has the largest overall effect of reducing variability in DNA replication speeds within a single cell (Extended Data Fig. 10a–d). In line with this, we observe that the variability in DNA replication speeds is higher in early S phase (WT DMSO) and is dramatically decreased upon addition of the inhibitor of RNA polymerase II DRB (Extended Data Fig. 10a–d). Moreover, elevated DNA damage (XRCC1Δ DMSO) increases the variability in DNA replication speeds compared with the steady-state (WT DMSO). The variability of replication speeds in both WT and XRCC1Δ cells are decreased following addition of the PARP inhibitor Olaparib (Extended Data Fig. 10a–d). In WT cells, this decrease seems to be concentrated in early S phase (Extended Data Fig. 10e). Conversely, XRCC1Δ cells display higher levels of variability throughout S phase, which are diminished throughout S phase with PARPi (Extended Data Fig. 10e). This indicates that it is not only transcription that impacts DNA replication speeds and variability within single cells. Nonetheless, these results imply that transcription-coupled damage increases the variability in speed between DNA replication forks in early S-phase cells. Furthermore, our findings suggest that PARP activity is critical in regulating DNA replication speeds in response to transcription-coupled DNA damage.

**Fig. 3 | Transcription limits DNA replication speeds in early S phase.**
**a**, Binned histogram of distances (kb, *x* axis) and range-scaled density (*y* axis) single-cell pair correlations (black line) with a fitted model (dashed red line) of representative early, middle and late S-phase (lower) RPE-1 cells labeled with the Δ*t* = 75 min scheme. **b**, DNA replication speed (*x* axis) over S phase (*y* axis) in RPE-1 (*n* = 326) treated with DMSO subjected to the Δ*t* = 75 min labeling scheme. Each dot is a cell, the line indicates a rolling-window median smooth and the ribbon the standard deviation. **c**, Sorted early (yellow) and sorted late (green) S-phase RPE-1 hTERT FUCCI cells superimposed on all detected single-cell events during FACS. **d**, Detected DNA fiber length analysis using DNA combing analysis from the indicated cell population labeled with IdU (20 min labeling, 100 fibers per replicate, *n* = 3, two-sided Student's *t*-test, *P* < 2.99 × 10⁻²³). The boxplot is defined by the median ± interquartile range (IQR) and whiskers are 1.5× IQR. **e**, Fraction of scEdU-seq domains covered by expressed genes (y-axis) over S-phase (x-axis) colored by expression level (stacked). **f**, Nascent RNA-sequencing from S-phase RPE-1 cells. Rolling-window smoothened normalized scEdU-seq coverage of genes (y-axis) over S-phase progression (x-axis) colored by expression level. **g**, Maximum normalized counts (*y* axis) of DNA replication speeds (*x* axis) inside or outside the indicated regions. Adjusted *P* values with two-sided *t*-test are: H3K27me3 (0.512, nonsignificant (NS)), H3K36me3 (3.0 × 10⁻³, ***), H3k9me3 (3.9 × 10⁻⁴, ****) and transcribed (1.2 × 10⁻⁴, ****). **h**, DNA replication speed over S phase in RPE-1 treated with DMSO (gray, *n* = 326) or DRB (cyan, *n* = 713) subjected to Δ*t* = 75 min labeling scheme (lower left), the line and ribbon indicate the rolling-window median standard deviation. Difference in DNA replication speeds between DMSO and DRB in kb min⁻¹ (*y* axis) over S-phase progression (*x* axis, upper left), marginal density (*x* axis) of DNA replication speed in kb min⁻¹ (*y* axis) colored for DMSO-treated (gray) or DRB-treated cells (cyan, lower right) and cumulative distribution speeds (upper right). **i**, Maximum normalized counts of speeds inside or outside the indicated regions for DMSO-treated or DRB-treated RPE-1. Adjusted *P* values with two-sided *t*-test are: DMSO-H3K36me3 (3.0 × 10⁻³, ***), DMSO-transcribed (1.2 × 10⁻⁴, ****), DRB-H3K36me3 (1.1 × 10⁻², **) and DRB-transcribed (0.398, NS). RFS, replication fork speed.

## Discussion

We developed a method to profile DNA replication forks and their speeds in single cells. We observe that DNA replication speeds accelerate during S phase. Reduced DNA replication speeds at the start of S phase occur in genomic regions with high levels of RNAPII transcription. Inhibition of RNAPII transcription increases DNA replication speeds at these locations. We find that inhibition of RNAPII results in both lower PARP activity and less DNA damage. We continue to show

that lowering PARP activity allows for higher DNA replication speeds, specifically in early S phase. In addition, the hyperactivation of PARP in RPE-1 cells lacking XRCC1 results in a genome-wide decrease in DNA replication speeds. We can reverse this decrease by lowering PARP activity, indicating a direct role for PARPs in regulating DNA replication speeds. Overall, this implies that transcription-coupled DNA damage increases PARP activity, which in turn reduces DNA replication speed.

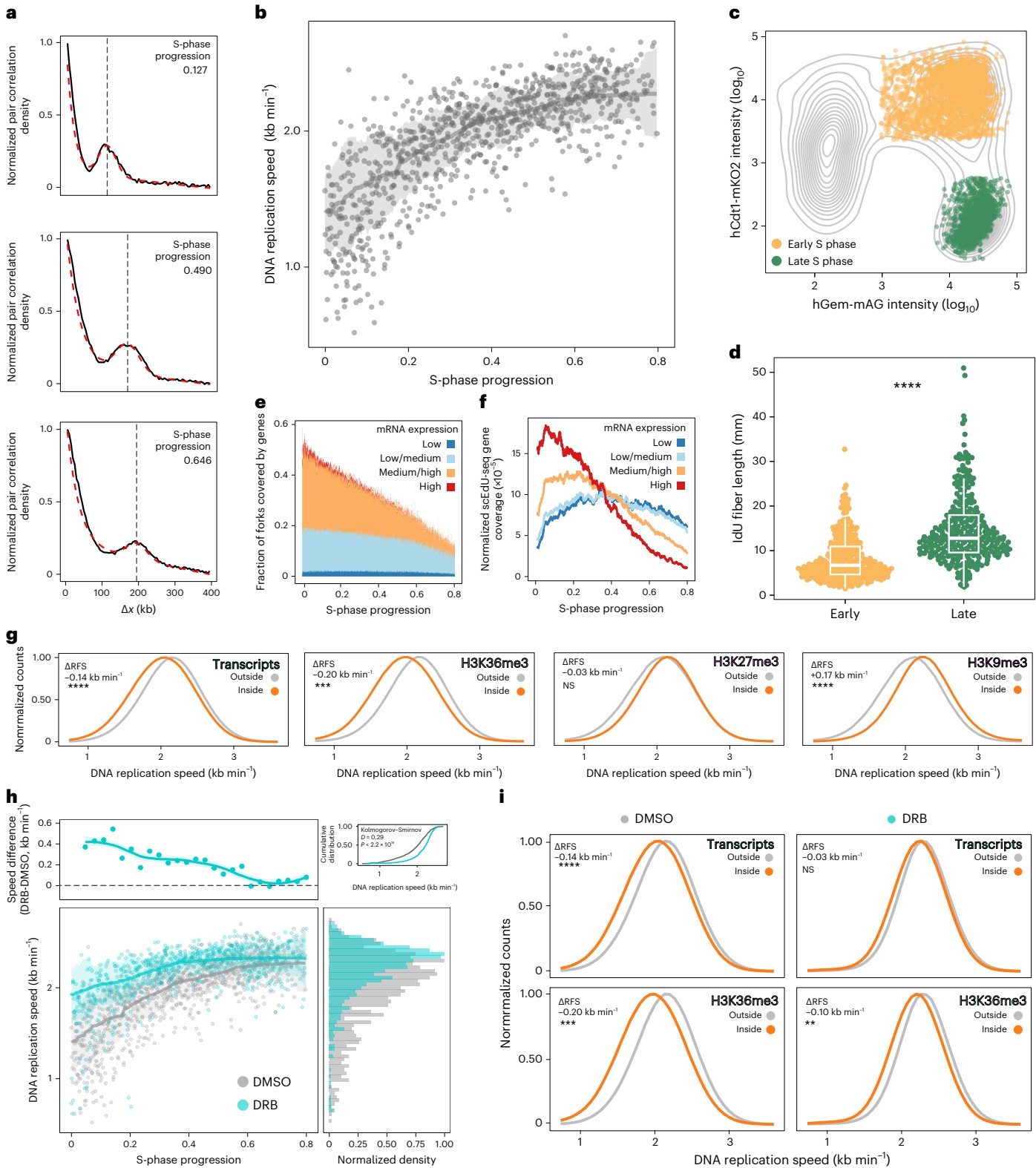

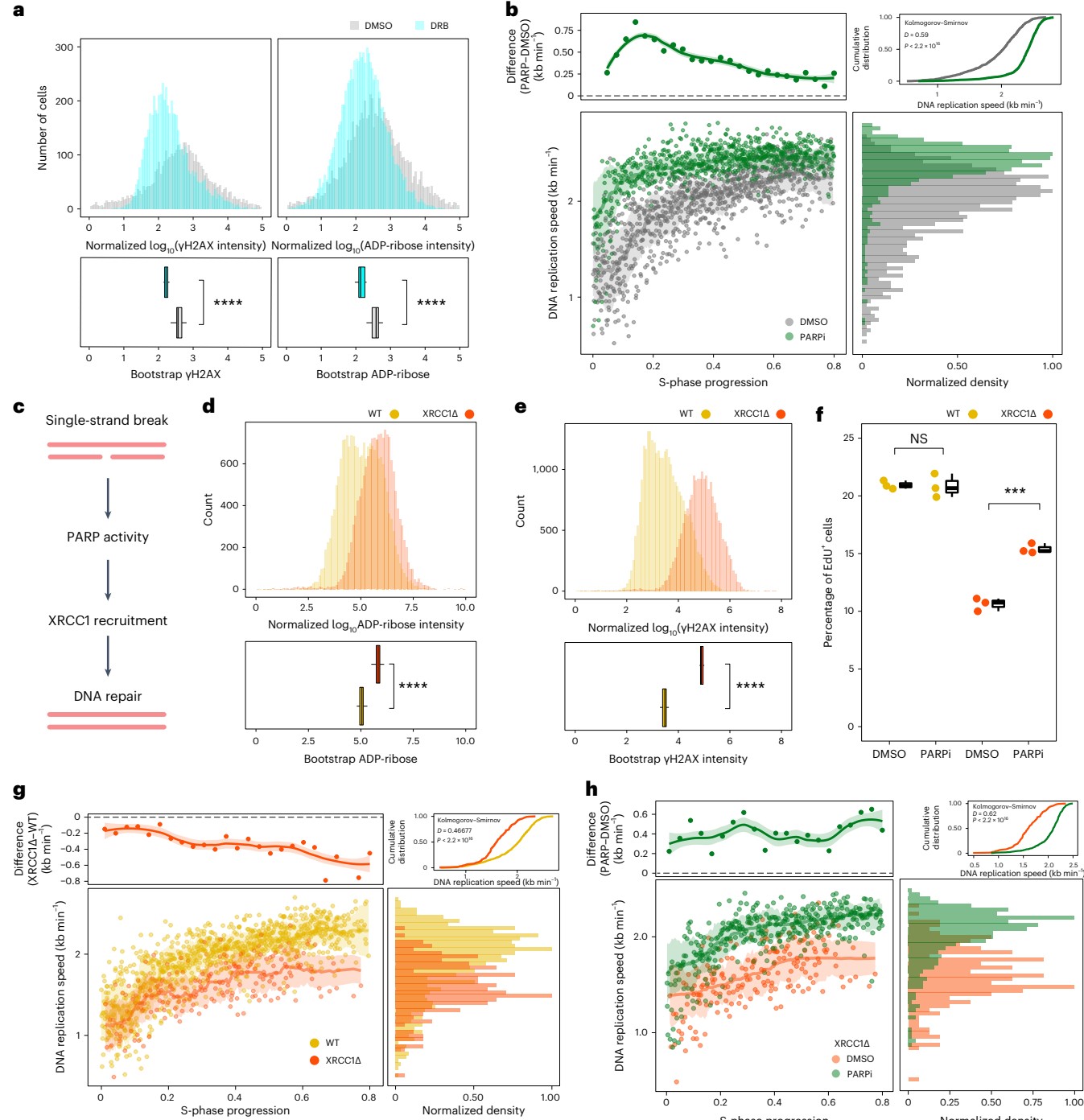

**Fig. 4 | Transcription-coupled damage decreases DNA replication speeds.**
**a**, Z-scored log$_{10}$-transformed ADP-ribose (left) or γH2AX (right) intensities for DMSO-treated (gray) or DRB-treated cells (cyan) and bootstrapped mean of intensities (lower). **b**, DNA replication speed over S phase in RPE-1 treated with DMSO (gray, n = 326) or PARPi (24 h, green, n = 766) (lower left) showing the difference in speeds between DMSO and PARP in kb min$^{-1}$ (y axis) over S phase (x axis, upper left), density (x axis) of DNA replication speeds in kb min$^{-1}$ (y axis) colored for DMSO- treated (gray) or PARP-treated cells (green) and cumulative frequency distribution of speeds (upper right). **c**, Cartoon of signaling and repair of single-strand breaks. **d**,**e**, Z-scored log$_{10}$-transformed ADP-ribose (**d**) or γH2AX (**e**) intensities (x axis) for WT (yellow) or XRCC1Δ RPE-1 cells (orange) and bootstrapped mean of intensities (lower). A two-sided Student's t-test with multiple testing correction (Bonferroni) was performed (n = 3; pan adenosine diphosphate ribose (pADPr), P < 2.81 × 10$^{-8}$; γH2AX, P < 3.05 × 10$^{-15}$). **f**, Percentage of EdU$^+$ cells (y axis) for DMSO versus PARPi (24 h) (x axis) colored by WT (yellow)

or XRCC1Δ (red) (n = 3; two-sided Student's t-test; ***P = 5.82 × 10$^{-4}$), boxplots are defined by the median ± IQR and whiskers are 1.5× IQR. **g**, DNA replication speed over S phase in WT (yellow, n = 326) or XRCC1Δ (red, n = 187) (lower left). Each dot is a cell, the line indicates a rolling-window median smooth and the ribbon the standard deviation. Difference in DNA replication speeds between WT and XRCC1Δ in kb min$^{-1}$ (y axis) over S phase (x axis, upper left). The histogram shows DNA replication speeds in kb min$^{-1}$ (y axis) colored for WT (yellow) or XRCC1Δ (red, lower right) and cumulative frequency distribution (upper right). **h**, DNA replication speed over S phase in XRCC1Δ treated with DMSO (red, n = 187) or PARP (4 h, green, n = 393) (lower left). Each dot is a cell, the line indicates a rolling-window median smooth and the ribbon the standard deviation. Difference in DNA replication speeds between DMSO and PARPi (4 h) in kb min$^{-1}$ (y axis) over S phase (x axis, upper left), marginal density (x axis) of DNA replication speeds in kb min$^{-1}$ (y axis) colored for DMSO-treated (red) or PARP-treated cells (green, lower right) and cumulative frequency distribution of speeds (upper right).

Our data suggest crosstalk between DNA replication fork speeds and transcription through the activity of PARP enzymes. Proteomic profiling has identified DNA replication and transcription as the two biological processes regulated through ADP-ribosylation by PARP enzymes[38]. Several replication factors are known to regulate fork speeds[25,39,40]. However, which specific components are regulated by PARPs to reduce replication fork speeds remains to be discovered. Outside transcribed regions, we have identified several other factors contributing to DNA replication speeds. Previous studies have implied that chromatin states control replication speeds by studying the inactive X-chromosome in hybrid mice[41]. These observations and previous findings provide an interesting avenue for future studies

There are several advantages to using the nucleotide analog EdU. scEdU-seq does not require internal normalization to sorted G1 cells. Therefore, we can extract DNA replication profiles from heterogeneous samples from highly divergent sources bearing different karyotypes, as evidenced by chromosomal gains 10q and 12 in RPE-1 hTERT cells (Extended Data Fig. 2g,h). This feature of scEdU-seq enables DNA replication research at the single-cell level in non-copy number variable material, which is of particular interest in samples with replication stress mutational signatures[42]. Alternatively, unscheduled DNA replication by a variety of DNA repair pathways could potentially be detected using scEdU-seq. Furthermore, reagent costs for scEdU-seq are lower than for methods based on whole-genome sequencing because only the nascent DNA is sequenced. In combination with smaller reaction volumes and a lack of reliance on commercial kits, this results in increased scalability. The scalability of scEdU-seq enables small-scale chemical/genetic screens to identify regulators of DNA replication. These screens might prove useful in uncovering the mechanisms of action of Olaparib (PARPi) on DNA replication speeds in transcribed regions of the genome. Finally, extracting transcriptome profiles in conjunction with DNA replication profiles will allow the uncovering of molecular crosstalk as well as the identification and characterization of rare DNA replication events.

In addition to using scEdU-seq to profile DNA replication speeds, long-read sequencing has been used to determine replication dynamics[11–13,26]. In terms of cost, scEdU-seq and nanopore-based long-read sequencing are comparable. Long-read sequencing has single base pair resolution, whereas scEdU-seq is limited by the distance between restriction enzyme digest sites in the genome (~250 bp for NlaIII). Therefore, scEdU-seq cannot quantify replication speed in small regions of the genome such as promoters, enhancers and R-loops (~0.2–2 kb in length)[35,43]. Nanopore-based methods are better suited for this; however, this technology is not yet applicable at the single-cell level. In addition, determining replication timing (S-phase progression) based solely on long-read sequencing is challenging. Moreover, only a fraction of the long reads are labeled with nucleotide analogs, which results in ~99% of reads being unlabeled[26]. Therefore, which method is better depends on the particular hypothesis. We feel that scEdU-seq and nanopore-based technology are highly complementary.

Another complementary technology, scRepli-Seq, enables the quantification of replication timing in single cells. scRepli-Seq can be performed on fixed materials because it does not require the incorporation of a synthetic uridine analog. In addition, scRepli-Seq can be performed at a higher throughput using droplet-based methods, easily profiling tenfold more cells compared with scEdU-seq. Finally, the availability of commercial kits for single-cell DNA sequencing enables increased accessibility of scRepli-Seq to the community[44,45]. Conversely, we feel that scEdU-seq has advantages over scRepli-Seq. As previously stated, there is no need for copy number correction in scEdU-seq. Furthermore, scEdU-seq reagent costs are lower owing to the smaller custom reactions per cell and because fewer sequencing reads are required per cell, resulting in roughly 50-fold lower costs for scEdU-seq compared with scRepli-Seq (Supplementary Table 1).

A potential limitation of scEdU-seq (and related techniques based on metabolic labeling of DNA) is that it relies on the incorporation of nonnatural nucleotides, which might induce a stress response or affect DNA repair[46–48]. Searching for endogenous read-outs for active DNA replication in single cells is important to circumvent such a potential limitation. In addition, scEdU-seq cannot be applied to bio-banked material and other situations in which nucleotide analog labeling is not feasible. Moreover, cells with low EdU incorporation rates (for example, cells at the very beginning of S phase) might be excluded during the quality control step, which could possibly be mitigated by longer labeling or increased enrichment by FACS. Of note for the double-pulse EdU-labeling experiments, we are not able to measure the speed of all active forks in the cell. For instance, when a fork is labeled during the first pulse and is annihilated by another fork or when an initiation site fires before the second EdU pulse, we are unable to detect DNA replication speeds. We hope that scEdU-seq will enable the identification of single-cell DNA replication dynamics as well as replication speeds in a wide range of biological systems.

## Online content

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

## Methods

### Cell lines and reagents
RPE-1 hTERT FUCCI, RPE-1 hTERT and RPE-1 hTERT XRCC1Δ cells were cultured in DMEM/F12 supplemented with 10% FBS (Gibco), 1× GlutaMAX (Gibco) and 1× Pen-Strep (Gibco) at 37 °C with 5% CO$_2$. Human-induced pluripotent stem cells were cultured on vitronectin-coated plates in Essential E8 medium supplemented with penicillin/streptomycin and Revitacell (first 24 h, ThermoFisher Scientific). RPE-1 cells routinely tested negative for *Mycoplasma* contamination and were not authenticated. Cell counting was performed with the Bio-Rad TC-20 Cell Counter. The following chemicals were used: 5-ethynyluridine (EU) (100 μM, Invitrogen), EdU (10 μM, Invitrogen), Olaparib (AZD2281, 10 μM, Cell Signaling Technology), DRB (10 μM, Sigma), DAPI (2 μg ml$^{-1}$, ThermoFisher Scientific) and SN-38 (at the indicated concentrations; SelleckChem)

### XRCC1Δ knockout generation in RPE-1 hTERT cells
Guide RNA was designed using CRISPOR against the second exon of XRCC1. The primers 5′-CACCGAGACACTTACCGAAAATGGC-3′ and 5′-AAACGCCATTTTCGGTAAGTGTCTC-3′ (Integrated DNA Technologies) were cloned into pX330 (ref. [49]). Cells were co-transfected with pDonor-Blast[50] and pX330-XRCC1, allowed to recover for 72 h and selected with blasticidin. Clones were picked and expanded. The picked clones were validated by western blot analysis as previously described[51] and probed with CDK4 (1:1,000; Santa-Cruz Biotechnologies, cat. no. sc-260) and XRCC1 (1:1,000; Abcam, cat. no. ab1838) primary antibodies. Secondary goat-anti-rabbit and goat-anti-mouse horseradish peroxidase-conjugated antibodies were used for detection on Bio-Rad Gel-Doc (1:2,000; DAKO).

### SN-38 proliferation assay
Some 500 RPE-1 hTERT or RPE-1 hTERT XRCC1Δ cells were plated in a 96-well plate and treated with increasing concentrations of SN-38 for 120 h. Cell viability was measured at the end of the experiment with a CellTiter-Glo cell viability assay according to the manufacturer's protocol.

### Single- and double-pulse EdU treatment
For single-pulse experiments, $1.5 \times 10^6$ cells were treated with 10 μM EdU for 15 min, trypsinized and washed with 1× PBS buffer, followed by fixation in 75% ice-cold ethanol. Double-pulse experiments were performed by treating $1.5 \times 10^6$ cells with 10 μM EdU for 15 min. Cells were subsequently washed three times with DMEM/F12 medium and allowed to recover for the indicated periods (for example, 30, 60 or 90 min). Finally, cells were treated with a second pulse of EdU (10 μM), trypsinized, washed in PBS and fixed in 4 ml of 75% ethanol.

Fixed cells were stored at −20 °C for up to 24 h. For longer storage periods of up to 3 months, cells were stored in 4 ml of storage buffer (150 mM NaCl, 20 mM HEPES, 2 mM EDTA, 25 mM Spermidine with 10% DMSO) at −20 °C.

### Azide−PEG3−biotin EdU-click reaction
Eppendorf Protein Lo-Bind 0.5-ml tubes were precoated with 0.25% BSA in PBS. Afterwards 500 μl of cells, in either 75% ethanol or storage buffer, were pelleted for 3 min at 600g. The cells were resuspended in 0.25% BSA in PBS and left to block for 30 min at 4 °C. Following blocking, cells were pelleted and the click reaction was performed in situ in 50-μl reactions using the EdU-Click 647 imaging kit (Invitrogen) according to the manufacturer's protocol with some alterations. Azide-647 was replaced with azide−PEG3−biotin conjugate (Sigma, 2 mM) and supplemented with 6 mM tris((1-hydroxy-propyl-1H-1,2,3-triazol-4-yl)methyl)amine (Jena Bioscience).

### FACS
Following the click reaction, RPE-1 cells were washed once in 1× PBS, resuspended in PBS with 0.25% BSA (ThermoFisher Scientific) and 10 μg ml$^{-1}$ DAPI, and passed through a 20-μm mesh. Single cells were index sorted into a 384-well plate using BD FACS Influx with the following settings: sort objective single cells, a drop envelope of 1.0 drop, a phase mask of 10/16, a maximum of 16 extra coincidence bits, a drop frequency of 38 kHz, a 100-μm nozzle with a pressure of 18 pounds per square inch and a flow rate of ~100 events per s, which results in a minimum sorting time of ~5 min per plate.

Doublets and debris were excluded by using the forward and side scatter and the DAPI channel. For the hTERT RPE-1 FUCCI cells, measurements in the DAPI channel were used to enrich S-phase cells. The intensities in the monomeric Azami-Green and monomeric Kusabira-Orange 2 as well as DAPI channels were acquired and later used for data analysis. Single cells were sorted into 384-well hardshell plates (Bio-Rad) containing 5 μl of light mineral oil (Sigma-Aldrich).

### Library preparation
Library construction progressed through three general steps (Fig. 1a). Reagents were dispensed to 384-microwell plates using either Nanodrop II (Innovadyne Technologies) or Mosquito (TTP Labtech). Plates were spun at 2,000g for 2 min after each liquid transfer step.

### Cell lysis and NlaIII digestion
After sorting, single cells were lysed in 100 nl of lysis mix (10 nl of 1× Cut-Smart buffer (NEB), 10 nl of proteinase K (Ambion), 80 nl of H$_2$O). Plates were incubated for 2 h at 55 °C and the proteinase K was heat-inactivated for 20 min at 80 °C. The genome was digested with 100 nl of NlaIII mix (10 nl of 1× CutSmart buffer (NEB), 10 nl of NlaIII (Ambion), 80 nl of nuclease-free H$_2$O) at 37 °C for 4 h and heat-inactivated for 30 min at 65 °C.

### End-repair and A-tailing followed by adapter ligation
To end-repair NlaIII overhang, we next incubated single cells with 100 nl of end-repair mix (1.6 nl of Klenow large fragment (NEB), 1.6 nl of T4 polynucleotide kinase reaction buffer (NEB), 4 nl of 10 mM deoxynucleotide triphosphates (dNTPs), 2.3 nl of 100 mM ATP, 6.6 nl of 25 mM MgCl$_2$, 5 nl of polyethylene glycol 8000 (PEG8000; 50%, NEB), 1.2 nl of 20 ng ml$^{-1}$ BSA (NEB), 23.3 nl of 10× polynucleotide kinase reaction buffer (NEB) and 54.2 nl of nuclease-free H$_2$O) for 30 min at 37 °C and heat-inactivated both enzymes for 20 min at 75 °C. To ligate adapters with a T-overhang, we A-tailed the end-repaired genomic DNA fragments with 100 nl of A-tailing mix (0.66 nl of AmpliTaq 360 (ThermoFisher Scientific), 0.66 nl of 100 mM dATP, 16.6 nl of 1 M KCl, 5 nl of PEG8000 (50%, NEB) 0.5 nl of BSA (20 ng/ml, NEB) and 77.2 nl of nuclease-free H$_2$O) for 15 min at 72 °C. Finally, A-tailed fragments were ligated to 50 nl of 5 mM T7 promoter containing adapters[15] with cell barcodes and UMI (Supplementary Table 2) using 150 nl of ligation mix (25 nl of T4 DNA ligase (400,000U, NEB), 3.5 nl of MgCl$_2$, 10.5 nl of Tris buffer pH 7.5 (1 M, Gibco), 5.25 nl of dithiothreitol (1 M, ThermoFisher Scientific), 3.5 nl of ATP (100 mM, ThermoFisher Scientific), 10 nl of PEG8000 (50%, NEB), 1 nl of BSA (20 ng ml$^{-1}$ NEB) and 91.25 nl of nuclease-free H$_2$O) for 20 min at 4 °C followed by 16 h at 16 °C and heat-inactivation for 20 min at 65 °C.

### Pooling and purification of EdU fragments
The contents of each plate were collected into VBLOK200 reservoirs precoated with mineral oil (ClickBio) by centrifuging at 300g for 1 min. The aqueous phase was collected and separated from any residual mineral oil by centrifugation. EdU−PEG3−biotin containing DNA molecules was affinity purified using MyOne Streptavidin C1 magnetic beads (Invitrogen) according to the manufacturer's protocol. Subsequently, we retrieved the complementary strand of the EdU−PEG3−biotin containing the DNA strand by heat denaturation at 95 °C. While ramping down the temperature (0.1 °C s$^{-1}$) to 20 °C, we annealed an oligo (5′-ATGCCGGTAATACGACTCAC-3′) complimentary to the constant adapter sequence region in oligo annealing buffer (20 mM Tris pH 8,

1 mM MgCl$_2$, 100 mM NaCl). Next, we extended the primer to generate double-strand DNA using Klenow large fragment mix (1× NEB Buffer 2, 50 mM dNTPs, 0.5 U of Klenow large fragment) for 45 min at 25 °C, and heat-inactivation for 20 min at 75 °C. DNA fragments were purified with Ampure XP beads (Beckman Coulter) at a sample to beads ratio 1:1 and resuspended in 7 µl of nuclease-free H$_2$O.

### Library amplification by in-vitro transcription and PCR

Preamplified libraries were linearly amplified using a MEGAscript T7 Transcription Kit (ThermoFisher Scientific) for 12 h at 37 °C. Template DNA was removed by the addition of 2 µl of TurboDNAse (ThermoFisher Scientific) for 15 min at 37 °C. Amplified RNA (aRNA) was fragmented for 2 min at 94 °C with fragmentation buffer (5× concentrated; 200 mM Tris-acetate pH 8.1, 500 mM KOAc, 150 mM MgOAc). aRNA was directly cooled to 4 °C on ice and 50 mM EDTA was added to stop fragmentation. The fragmented RNA is purified using RNA Clean XP beads (Beckman Coulter) at a beads to sample ratio of 1:1, and eluted in 12 µl of H$_2$O. Next, 5 µl of aRNA was converted to cDNA by RT in two steps. First, the RNA was primed for RT by adding 0.5 µl of dNTPs (10 mM) and 1 µl of random hexamer RT primer 20 µM (5′-GCCTTGGCACCCGAGAATTCCANNNNNN-3′) at 65 °C for 5 min followed by direct cooling on ice. Second, RT was performed by the addition of 2 µl of first-strand buffer, 1 µl of 0.1 M dithiothreitol, 0.5 µl of RNAseOUT and 0.5 µl of Superscript II, and incubating the mixture at 25 °C for 10 min, followed by 60 min at 42 °C and 20 min at 70 °C. Single-strand cDNA was purified from aRNA through incubation with 0.5 µl of RNAseA (ThermoFisher Scientific) for 30 min at 37 °C. Finally, cDNA was amplified by PCR, which also attaches the Illumina small RNA barcodes and handles (Supplementary Table 3), by adding 25 µl of NEBNext Ultra II Q5 Master Mix (NEB), 11 µl of H$_2$O and 2 µl of RP1 and RPIx primers (10 µM). DNA fragments were purified twice with Ampure XP beads (Beckman Coulter) at a sample to beads ratio 0.8:1, and were resuspended in 10 µl of nuclease-free H$_2$O. The abundance and quality of the final library were assessed by Qubit and Bioanalyzer.

### Sequencing

Libraries were sequenced using v2.5 chemistry on a NextSeq500 or NextSeq2000 (Illumina; NextSeq control software v.2.2.0.4; RTA v.2.4.11) with 100 cycles for read 1 (cell index and UMI) and 100 cycles for read 2 (sample index).

### DNA fiber assay

RPE-1 hTERT FUCCI cells were labeled with 250 µM 5-iodo-2-deoxyuridine (IdU) for 20 min and then chased with 2 mM thymidine. Following trypsinization, cells were resuspended in 0.2% BSA–PBS0 and sorted for specified S-phase fractions. Subsequently, RPE-1 hTERT cells were lysed on microscopy slides in lysis buffer (0.5% SDS, 200 mM Tris pH 7.4, 50 mM EDTA). DNA fibers were spread by tilting the slide and were subsequently air dried and fixed in methanol/acetic acid (3:1) for 10 min. For immunolabeling, spreads were treated with 2.5 M HCl for 90 min. IdU was detected by staining with mouse-anti-BrdU (1:250; BD Biosciences, cat. no. 347580) for 1 h and was further incubated with Alexa Fluor 647-conjugated anti-mouse immunoglobulin G (1:500) for 90 min. Images were acquired on a Zeiss Axio Imager Z2 and fiber lengths were scored using ImageJ.

### Flow cytometry

Apoptosis analysis was performed using an Annexin-V-APC kit (BioLegend) as described in the manufacturer's protocol for all treatment conditions and cell lines. Cell-cycle analysis was performed by fixing RPE-1 hTERT FUCCI, RPE-1 hTERT or RPE-1 hTERT XRCC1Δ cells in 70% ethanol and counterstaining with DAPI (10 µg ml$^{-1}$, 20 min on ice). For nascent RNA labeling, we treated cells with EU (200 µM) for 1 h and fixed the cells in 75% ethanol. For DNA replication labeling, we incubated cells

for 30 min with 10 µM EdU. For both EU and EdU labeling, we used the EdU-Click 647 Imaging Kit according to the manufacturer's protocol. For γH2AX staining, we used the fluorescein isothiocyanate-conjugated γH2AX (1:500, Millipore). panADP-ribose binding reagent (1:1,500; Sigma-Aldrich, cat. no. MABE1016) was used in combination with donkey anti-rabbit immunoglobulin G secondary antibody with different Alexa Fluor conjugations depending on the experimental conditions and cell line used.

### Read processing

Processing of raw fastq to count tables was performed using SingleCellMultiOmics v.0.1.25 (https://github.com/BuysDB/SingleCellMultiOmics).

First, fastq files were demultiplexed, which adds UMI, cell, sample and sequencing indices to the header of the fastq. Cell barcodes and UMIs with a hamming distance of 1 were collapsed. Next, the adapter sequences were trimmed from each read with cutadapt. Subsequently, reads were mapped with BWA using the mem function to Ensembl release 97, GRCh38.p12 for *Homo sapiens*, the bam outputs were sorted with samtools. Mapped reads were subjected to molecule assignment, which generates tags for NlaIII restriction site position and integrates the cell barcode, UMI, library, strand and genomic position of NlaIII restriction site into one tag. This integrated molecule tag allows for deduplication of reads and the generation of long-format tables. These tables were filtered for the presence of a NlaIII restriction site, a mapping quality >30, the molecule has a pair of reads assigned, the molecule is unique and should not have alternative alignment positions in the genome.

### Single-cell DNA replication analyses and plotting

All data analysis was done in R using the tidyverse and data.table packages unless otherwise stated.

### S-phase ordering

First, cells were filtered by the average counts per 100 kb bin with a lower threshold (single pulse 0.37; double pulse 0.08) and an upper threshold (single pulse 2.72; double pulse 12.18), and deviance of Poisson behavior defined as:

$$\log(\text{Coefficient of Variation}) > -0.5 \times \log(\text{mean}) + \text{threshold}$$

where the threshold was set to 0.1. An exponential mixture model was fitted on the distances between successive reads for each single cell separately using the R package flexmix. Subsequently, reads with a posterior probability of >0.5 for the distances to their first neighbors were used for S-phase ordering. Next, we performed a gaussian kernel smoothing (s.d. of 8333.333) for the remaining reads, after which the pairwise overlap coefficient was calculated between all cells on a per chromosome basis. The overlap coefficient was converted to a distance as follows:

$$\text{Distance} = \min\left(\frac{1}{\text{score}} - 1,\ 1000\right)$$

and averaged over chromosomes. The resulting distance was embedded in one dimension using UMAP implemented by the R package umap. This UMAP computation was repeated 100 times and the resulting UMAP axis was converted to a z-score. To prevent flipping of the direction of S-phase progression, runs that had an average Spearman rank correlation of <0.85 with the other runs were discarded. Finally, to determine whether a cell could be placed on the ordering definitively, placings were considered clustered if the smallest distance between two successive placings was <0.1, and cells were kept if the biggest cluster contained at least 80% of the successful runs.

## HMM pulse segmentation

To determine the part of the genome that is undergoing replication in a single-pulse experiment, a two-state HMM was used to segment the genome into foreground and background. We used the R package mhsmm[52] to fit a hidden semi-Markov model with exponential emission distributions and a gamma sojourn distribution per cell on the distances between neighboring reads (Extended Data Fig. 2a), where each of the chromosomes was used as a separate observation. Reads generated by the same polymerase are expected to be close together on the genome, closer in general than neighboring forks or spurious background reads. Subsequent distances between reads are assigned to the foreground state (Extended Data Fig. 2a, yellow) and background (Extended Data Fig. 2a, red). To get the most likely sequence of states, the viterbi algorithm implemented in the mhsmm package was used then. Thus, reads that were generated by the same polymerase have a higher posterior probability (extracted from the viterbi algorithm, Extended Data Fig. 2b) of being close together on the genome, closer in general than neighboring forks or spurious background reads. Subsequent reads that are assigned to the foreground state by the model are considered a track traveled by a single polymerase. We found that DNA replication fork tracks contain, on average, ~7 scEdU-seq reads (Extended Data Fig. 2c) and the coverage is around 5% of the width of these DNA replication tracks (Extended Data Fig. 2d). Finally, we found that our DNA replication fork calls are, on average, 29,000 bp in length for a 15-min pulse of EdU in RPE-1 TERT cells (Extended Data Fig. 2d) and the majority of DNA replication forks have three or more reads per single cell (Extended Data Fig. 2f).

However, there are caveats with regards to the quantification of the number of forks per cell, which includes undercounting (that is, false negatives) that arise from low sampling of reads from a labeled stretch of EdU. Furthermore, the single-pulse labeling strategy used in this work (15 min of EdU) has a lower limit resolution for the DNA replication tracks (~15–20 kb). Several other methods such as Okazaki fragments sequencing (OK-seq) and fiber analyses enable more fine-grained analysis of processes occurring at higher resolution. For instance, within a single initiation zone (average ~30 kb) multiple initiation sites fire[53]. Using scEdU-seq, we cannot resolve these individual initiation sites using the current labeling schemes, library preparation and analysis. Finally, several DNA repair processes are known to fill in considerable stretches of DNA, which potentially result in false-positive DNA replication tracks[54].

## Comparison between Repli-Seq and bulk scEdU-seq

Bulk scEdU-seq samples were generated by collecting 500 early or late S-phase RPE-1 hTERT FUCCI cells treated with EdU for 120 min. Cells were processed similarly to scEdU-seq libraries. We retrieved bulk Repli-Seq for RPE-1 hTERT cells from the 4D Nucleome program, which was generated by the Gilbert lab. Subsequently, the samples were binned with a 50-kb resolution and reads per bin were z-scored per sample. Z-scores were used for comparative plotting of traces as well as computing the Spearman correlation between samples.

## Comparison between Repli-Seq and scEdU-seq

Segmented pulses ('HMM pulse segmentation') from the single-pulse dataset were overlapped with 10-kb bins from the Repli-Seq dataset using the foverlaps function from data.table. The Repli-Seq RT was calculated as follows:

$$\text{RT score} = \frac{\text{early} - \text{late}}{\text{early} + \text{late}}$$

The Pearson correlation was then calculated between binned S-phase progression and the Repli-Seq RT score.

## EdUseq-HU comparison

Raw early DNA replication origin data (EdUseq-HU) for RPE-1 hTERT was used from ref. [19] (BioProject PRJNA397123). Raw FASTQ files were trimmed and mapped to Ensembl release 97, GRCh38.p12 with BWA. To compare scEdU-seq and EdUseq-HU both datasets were binned at 100-kb resolution. Next, the cumulative number of reads over S-phase progression per bin was calculated, and was used to calculate the binwise Pearson correlation per cumulative reads over S phase.

## DNA replication track heatmaps

To visualize DNA replication tracks, we used a heatmap to plot scEdU-seq signal per single cell over S-phase progression. First, we ordered single cells based on their S-phase progression ('S-phase ordering'). Next, we selected a segment of the genome for visualization of DNA replication tracks containing both early and late replication domains (for example, chromosome 2 from 10 to 50 Mb). Subsequently, we normalized the scEdU-seq signal in 50-kb bins to the maximum value observed for that chromosome (that is, 0–1).

## Simulation of single- versus double-pulse speed quantification

The number of reads per pulse was drawn from a Poisson distribution of varying intensity to simulate average sampling depth per replication track. Subsequently, these reads were placed at locations from a uniform distribution within a replication track (that is, 0 or 1 for a single pulse and 2 or 3 for a double pulse). Placement of the reads by the uniform distribution resulted in a ground truth DNA replication speed of 1 kb min$^{-1}$ for both a single and a double pulse. One thousand pulses were generated per intensity value. Finally, the DNA replication speed was estimated (Extended Data Fig. 3b,c) for the single-pulse speed (Methods 'Single-pulse DNA replication speed estimate') and double pulse (the average of the pairwise distances between the first and second pulse).

## Pair correlation

The pair correlation was calculated as the pairwise distances between all reads in one cell per chromosome. For display, the count was calculated per 5-kb bin, distances >400 kb (except Extended Data Fig. 5 for which the max was 1 Mb) were discarded and the total counts were either sum normalized or range-scaled between 0 and 1.

## Single-pulse DNA replication speed estimate

The width of the HMM-segmented pulses ('HMM pulse segmentation') can be calculated from the genomic coordinates of the first and last read in a pulse. However, because of sampling, this width is likely to be an underestimation of the actual traveled path of the polymerase. The pulse width was corrected for sampling similarly to the estimation of the maximum of a sampled uniform distribution, as follows:

$$\text{Pulse width} = w + w/(n - 1)$$

Where $w$ is the genomic distance between the first and the last read in the pulse and $n$ is the number of reads in the pulse.

## Mixture model fits

A mixture model with four components (uniform, exponential, halve-normal and normal distribution) was fitted per cell using a custom expectation maximization algorithm with soft labels written in C++ and implemented using the R package Rcpp[55]. In the maximization step the parameters of the component distributions were updated using the weighted mean for the exponential and normal distribution, and the weighted variance for the half-normal and normal distribution. In addition, the mean of the exponential component was restricted to >1,000, and the previous probability of the exponential component was restricted to >0.01. The algorithm was run until a relative tolerance of $10^{-8}$ or a maximum of 100 iterations was reached.

## Pair correlation simulation

In the case of sampling with equal intensity, to simulate the pair correlation, first the read locations were drawn from a uniform distribution with a minimum of −200 and a maximum of 200, and the number of reads was drawn from a Poisson distribution with a mean of 80. Reads, $r$, were then retained with the following logic:

$$-45 \times s < r < -30 \times s \wedge 30 \times s < r < 45 \times s$$

where the scaling $s$ of the window was drawn from a truncated (at 0) normal distribution with a given mean and variance.

$$s \sim N(\mu, \sigma | x > 0)$$

In the case of unequal sampling intensity, first the double window was defined and scaled with a speed factor drawn from a normal distribution (truncated at 0) with a given mean and variance. The read locations were then drawn from a uniform distribution where the minimum and maximum were scaled according to the speed factor, and the number of reads was again drawn from a Poisson distribution with a mean of 80. Reads falling outside the earlier defined window were again discarded. For both scenarios this was done 2,000 times for every combination of mean and variance of the ground truth speed distribution, after which the pair correlation was calculated as described previously. Subsequently, fitted speed estimates were corrected for Poisson sampling artifacts by fitting a locally estimated scatterplot smoothing smoothing surface to the simulated data. After which, we predict the input mean or standard deviation parameter using the output mean and standard deviation. Finally, this model predicts the ground truth mean and standard deviation from the measured experimental mean and standard deviation.

## scRNA-seq data analysis

Count tables from ref. [31] were filtered for the EU-labeled fraction of messenger RNA, and cells were assigned to S phase if the cell-cycle progression score was >0.333 or <0.75. The total counts per gene for the S-phase pseudo-bulk were then transformed by adding 1, $\log_{10}$-transformed and rounded.

$$\text{Transformed count} = \text{round}(\log_{10}(\text{counts} + 1))$$

The genomic location was added to the genes using the hg38 Ensembl release 106. In the case of overlapping genes, the one with the higher count was given priority for the overlapping portion. Segmented pulses from the 15-min EdU dataset were overlapped with the expressed genes using the foverlaps function from data.table.

## Chromatin immunocleavage sequencing

RPE-1 hTERT cells were processed similarly to in ref. [15]. The following antibodies were used: anti-H3K9Me3 (1:100; Abcam, cat. no. ab8898), anti-H3K27Me3 (1:200; Cell Signaling Technologies, cat. no. C36B11) and anti-H3K36Me3 (1:2,000; ThermoFisher Scientific, cat. no. MA5-24687).

## DNA replication speed comparison

Pair correlation distances from $\Delta t$ = 75 min were overlapped with expressed genes from single cell 5-ethynyl-uridine sequencing (scEU-seq) on RPE-1 cells using the foverlaps function from data.table. Pair correlation distances were split between transcribed and nontranscribed regions and given weights depending on the fraction of distance overlap with the specific gene. Subsequently, these distances were weighted by the posterior estimate for the speed component, which we derived from the pair correlation mixture model. Finally, we plotted the maximum normalized density of the weighted speed derived distances for transcribed and nontranscribed regions.

Statistical testing was performed by resampling from the density distribution and significance was determined by Student's $t$-test.

## Reporting summary

Further information on research design is available in the Nature Portfolio Reporting Summary linked to this article.

## Data availability

Raw sequencing data, metadata and count tables have been made available in the Gene Expression Omnibus under the accession number GSE211037. Data for comparisons to scEU-seq was downloaded from Gene Expression Omnibus accessions GSE128365. Raw sequencing data of EdUseq-HU was downloaded from SRA (PRJNA397123). Data for replication timing was downloaded from the 4D Nucleome project (4DNBSKYMY5XL).

## Code availability

All scripts to process raw data and generate figures are available at https://github.com/vincentvbatenburg/scEdU-seq. A shiny web application for scEdU-seq replication tracks is available at https://sceduseq.eu/.

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

## Acknowledgements

We thank R. van der Linden, A. Pfauth and M. Everts for assistance during the experiments. We thank P. Knipscheer, J. Garaycoechea and F. Mattiroli for providing valuable feedback on the manuscript. We also thank all members of the van Oudenaarden lab for scientific discussions. This work was supported by a European Research Council Advanced grant (ERC-AdG grant no. 101053581-scTranslatomics), Nederlandse Organisatie voor Wetenschappelijk Onderzoek (NWO) TOP award (NWO-CW grant no. 714.016.001) and (NWO) VICI award (grant no. 09150182110019). This work is part of the Oncode Institute, which is partly financed by the Dutch Cancer Society. In addition, we thank the Hubrecht Sorting Facility and the Utrecht Sequencing Facility, subsidized by the University Medical Center Utrecht, the Hubrecht Institute, Utrecht University and The Netherlands X-omics Initiative (NWO project 184.034.019).

## Author contributions

J.v.d.B. and A.v.O. conceived and designed the project. J.v.d.B. developed the experimental scEdU-seq protocol and performed experiments. C.G. performed the chromatin immunocleavage sequencing experiments. R.B.T. and M.A.T.M.v.V. performed

DNA fiber assay and fiber analysis. A.d.J.-S. and S.P.A. cultured and EdU-labeled the human-induced pluripotent stem cells. V.v.B. developed the statistical and analytical framework to analyze scEdU-seq data with the help of J.v.d.B. and A.v.O. J.v.d.B., V.v.B. and A.v.O analyzed the data. J.v.d.B., V.v.B. and A.v.O. discussed and interpreted the results. J.v.d.B. wrote the manuscript with feedback from A.v.O. and V.v.B.

## Competing interests

The authors declare no competing interests.

## Additional information

**Extended data** is available for this paper at

**Correspondence and requests for materials** should be addressed to Jeroen van den Berg or Alexander van Oudenaarden.

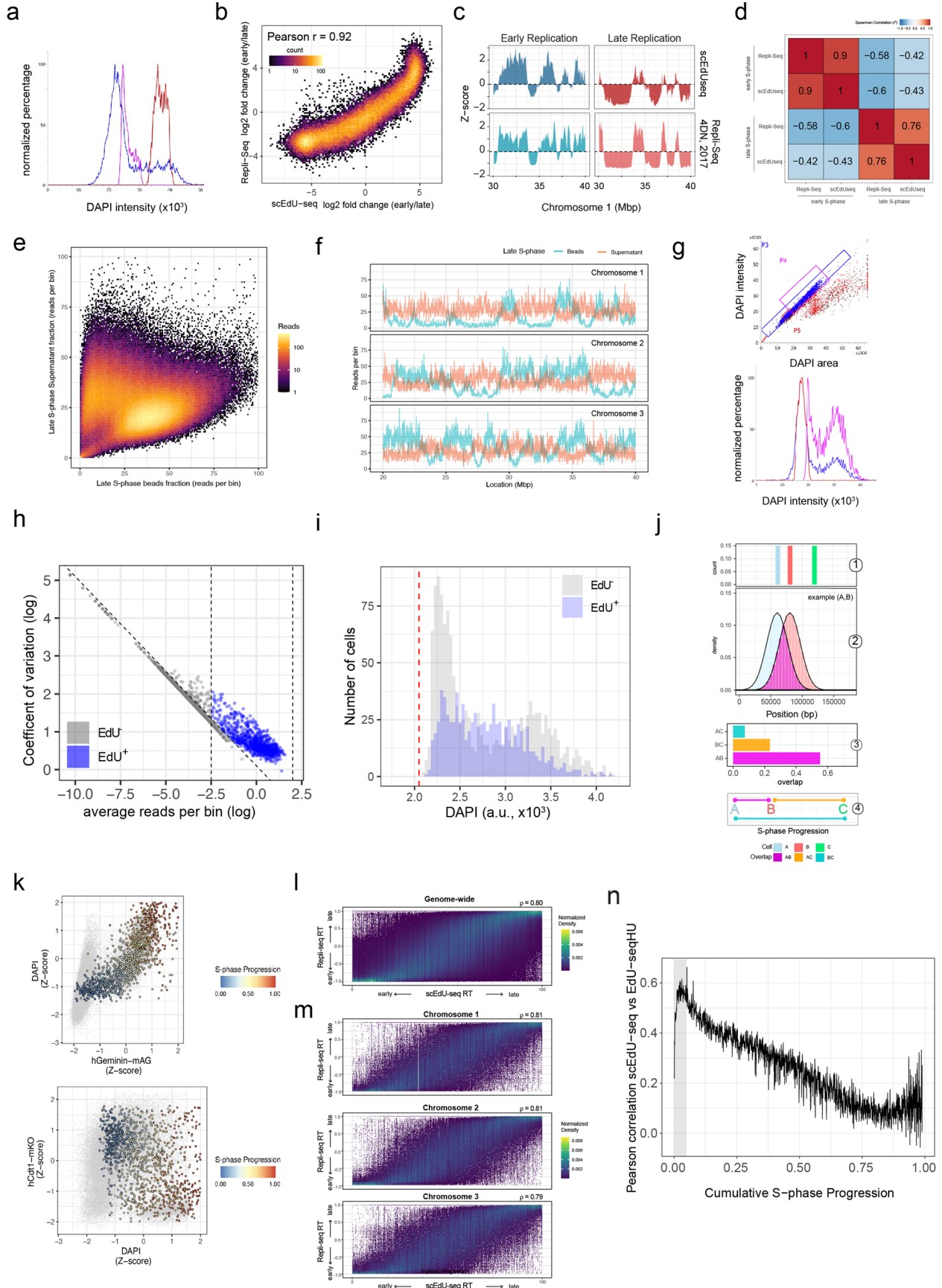

**Extended Data Fig. 1 | See next page for caption.**

**Extended Data Fig. 1 | Validation of scEdU-seq. a**. Enrichment gates for DNA content (DAPI) to sort early (*purple*) and late (*brown*) superimposed over the cell cycle distribution of cycling RPE-1 cells (*blue*). **b**. Scatter plot and Pearson correlation log2(early/late) for Repli-Seq (y-axis, 400.000 cells) and scEdU-seq (x-axis, 500 cells) with 120 minutes EdU treatment for each 50 kb in the genome **c**. Z-scored genome-coverage tracks of early (left) and late (right) S-phase samples for both scEdU-seq (500 cell bulk, top) and Repli-Seq (400k cell bulk, bottom) treated with 120 min. of EdU. **d**. Spearman rank correlation heatmap comparing early vs late S-phase sorted samples between Repli-Seq (400.000 cells) and scEdU-seq (500 cells) with 120 minutes EdU treatment**. e**. Scatter plot of 5 kb bins genome-wide with corresponding reads from supernatant (y-axis) or beads (x-axis) fraction from a Late S-phase RPE-1 bulk scEdU-seq sample. Density indicates the occurrences of read combination from beads and supernatant fraction. **f**. Read depth of beads (blue) or supernatant (red) fraction from Late S–phase RPE-1 bulk scEdU-seq. **g**. Flow cytometry gates for scEdU-sq experiments. Top - Doublets exclusion based on DAPI area and DAPI intensity for all single cells (blue P2) and S-phase enriched cells (purple, P4). Single cells for scEdU-seq were sorted from the P4 gate. Bottom - DNA content (DAPI) from single cells (P4, purple, scEdU-seq cells) superimposed over the cell cycle distribution of cycling RPE-1 cells (P2, blue) treated with 15 minutes EdU treatment.

**h**. Coefficient of Variation (y-axis) versus average reads per bin (x-axis) for all single pulse scEdU-seq cells. Each dot is a single cell and the top area between three dashed lines contain selected cells for subsequent analysis. **I**. Histogram of DAPI intensities for both scEdU-seq positive (blue) and negative (grey) cells. The red dashed line indicates the lower threshold of DAPI intensity used for cell sorting. **j**. Schematic representation computing S-phase progression. *1* - example distribution of reads from 3 different single cells A, B and C. *2* - Example of gaussian kernel smoothing of the reads from single cells A and B. *3* - Computing pairwise overlap coefficient between all cells. *4* - Converting overlap into distance metric between single cells. **k**. Scatter plot showing FUCCI reporters versus DNA content. Dots are single cells pseudo colored by the S-phase progression based on scEdU-seq tracks and in gray the cell cycle distribution of cycling RPE-1 cells. **l**. Genome-wide Repli-seq replication timing (y-axis) compared to S-phase progression determined by scEdU-seq (x-axis), color scale indicating normalized density of genomic bins **m**. Repli-seq replication timing (y-axis) compared to S-phase progression determined by scEdU-seq (x-axis), color scale indicating normalized density of genomic bins for chromosome 1 to 3, top to bottom respectively. **n**. Binwise Pearson correlation (y-axis) over cumulative S-phase progression (x-axis) between EdUseq-HU and scEdU-seq where the individual cells were aggregated up to the corresponding S-phase progression.

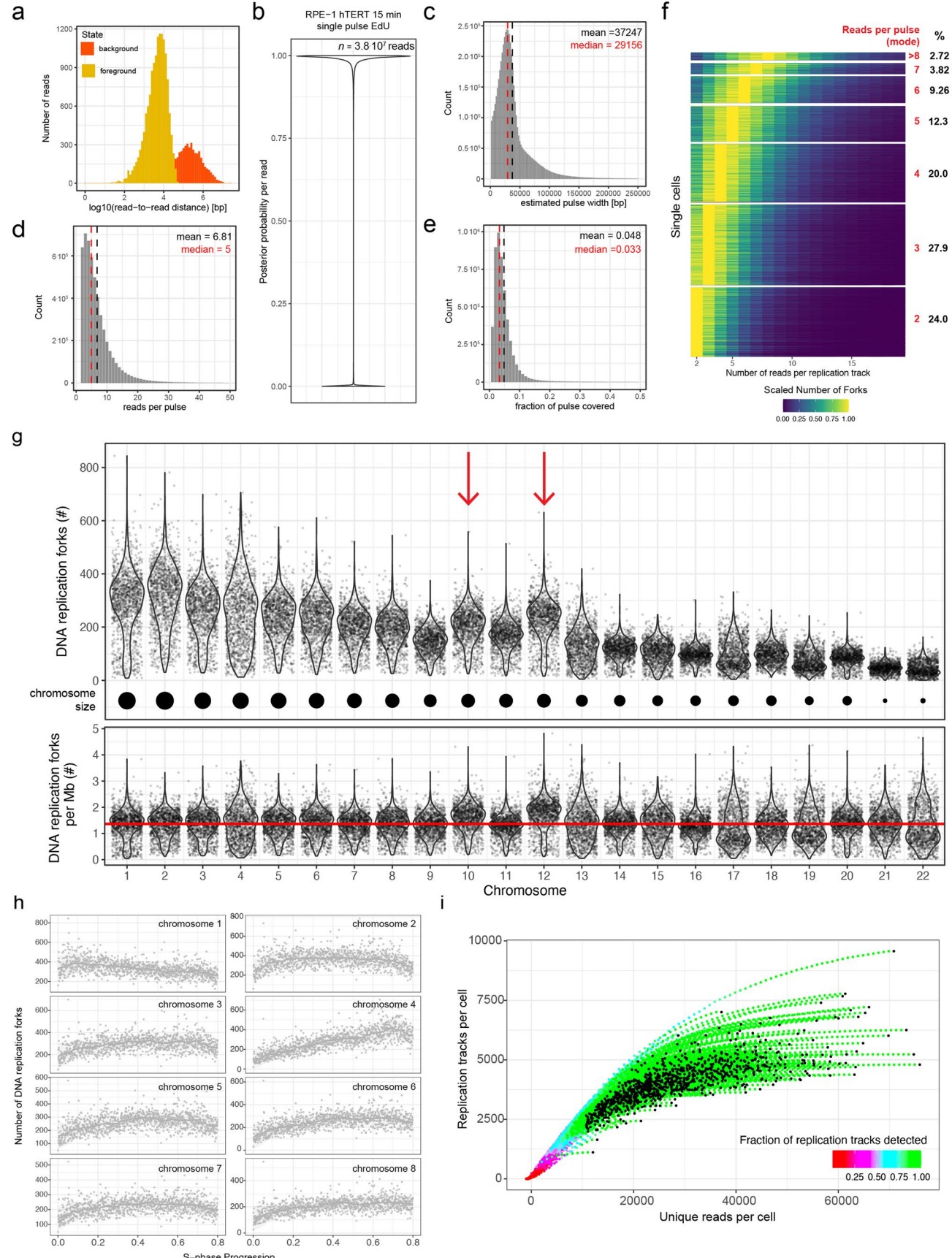

**Extended Data Fig. 2 | See next page for caption.**

**Extended Data Fig. 2 | Quantifying the number of replication tracks per cell.**
**a.** Count (y-axis) of read-to-read distance (y-axis), colored according to within pulse (yellow) or between pulses (red). **b.** Violin plot of posterior probability derived from HMM and viterbi algorithm, for each read for RPE-1 hTERT cells treated with 15 minutes of EdU and subjected to scEdU-seq. **c.** Count (y-axis) of the number of reads per segmented pulse (x-axis), dashed black line indicates the sample mean, dashed red line indicates the interpolated sample median. **d.** Count (y-axis) of the segmented pulse width (x-axis), dashed black line indicates the sample mean, dashed red line indicates the interpolated sample median. **e.** Count (y-axis) of fraction of genome covered per pulse (x-axis), dashed black line indicates the sample mean, dashed red line indicates the interpolated sample median. **f.** Heat-map displaying the number of sequenced reads per HMM-segmented replication track (x-axis) per single RPE-1 cell (y-axis) labeled with a single 15 min EdU pulse. The color-scale indicates the maximum-scaled number of replication tracks. The plot is split by the mode of reads per fork per single cells and cells are ordered by the ranked average reads-per-replication-track. **g.** Count (y-axis) of the number of forks per cell (x-axis). Red arrows indicate chromosomes with known whole (chr12) and partial (chr10q) chromosome gains in RPE-1 hTERT cells. **h.** Number of forks (y-axis) per cell versus S-phase progression (x-axis), for chromosome 1 to 8. **i.** Number of replication forks detected per cell (y-axis) versus the number of reads per cell (x-axis) where the cells were down-sampled by 10,000 reads. Each black dot is the original cell where the colored dots are the increasingly down-sampled cell colored by fraction of replication tracks recovered.

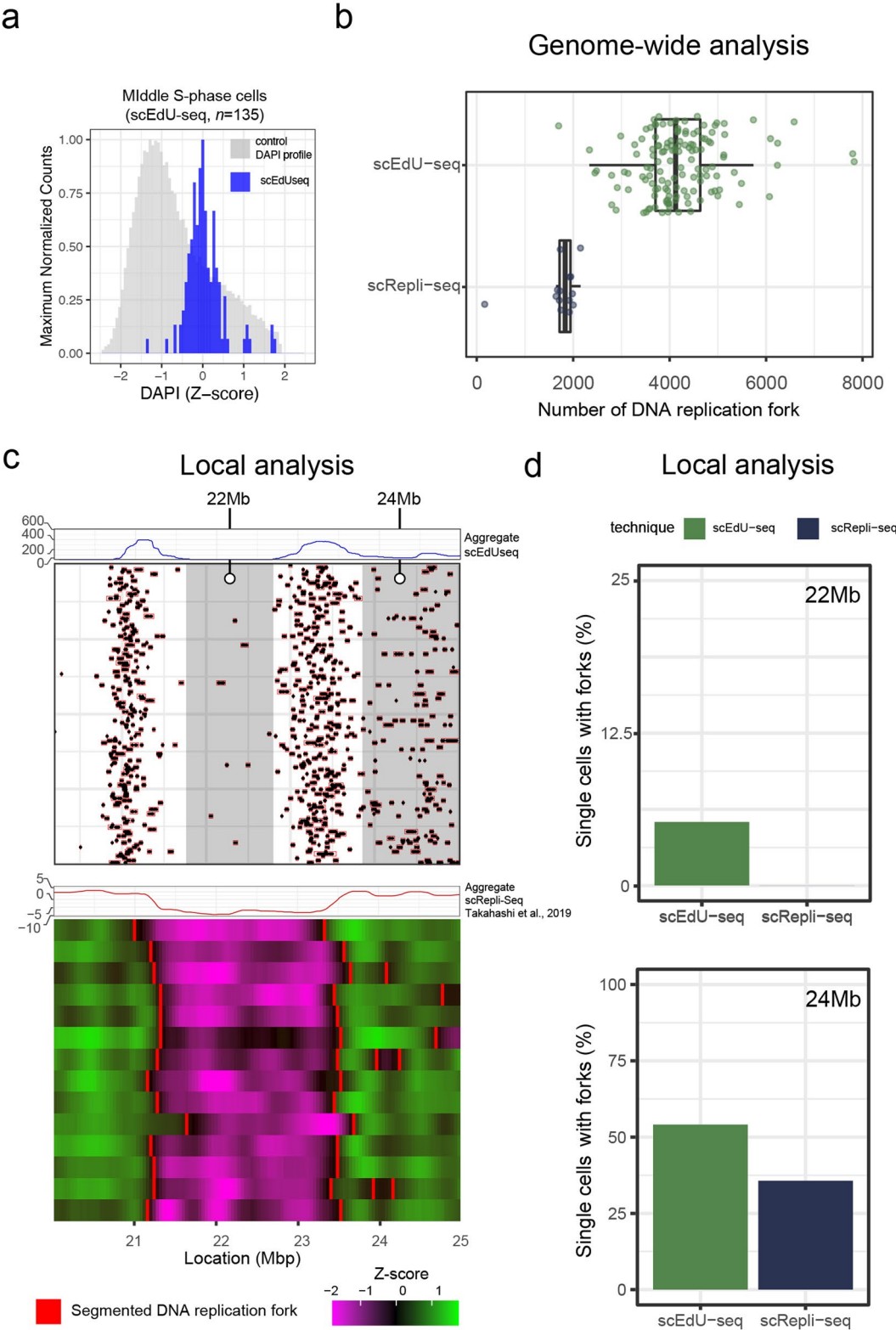

**Extended Data Fig. 3 | Comparing scEdU-seq to scRepli-seq. a.** Mid S-phase cells from scEdU-seq single pulse experiment for scRepli-seq comparison. DNA content (DAPI) intensities for Mid S-phase cells (blue) superimposed over cell cycle distribution of cycling RPE-1 cells (gray). **b.** Genome-wide DNA replication forks calls of single cells assayed by scRepli-seq (n = 14 from 1 replicate) and scEdU-seq (n = 135 cells from 1 replicate). The box of the boxplot is defined by the median ± IQR and the whiskers are 1.5X IQR. **c.** Heatmap of scEdUseq (15 min. single EdU pulse) maximum normalized log counts for middle S-phase cells binned per 5 kb (n = 135, bottom) and summed profile (top). Heatmap of scRepli-Seq log2 median counts of middle S-phase cells binned per 40 kb (n = 14, bottom) and summed profile (top). **d.** Quantification of the percentage of cells containing forks in indicated highlighted areas (chr2:22 Mb & chr2:24 Mb) for scEdU-seq (green) and scRepli-seq (blue).

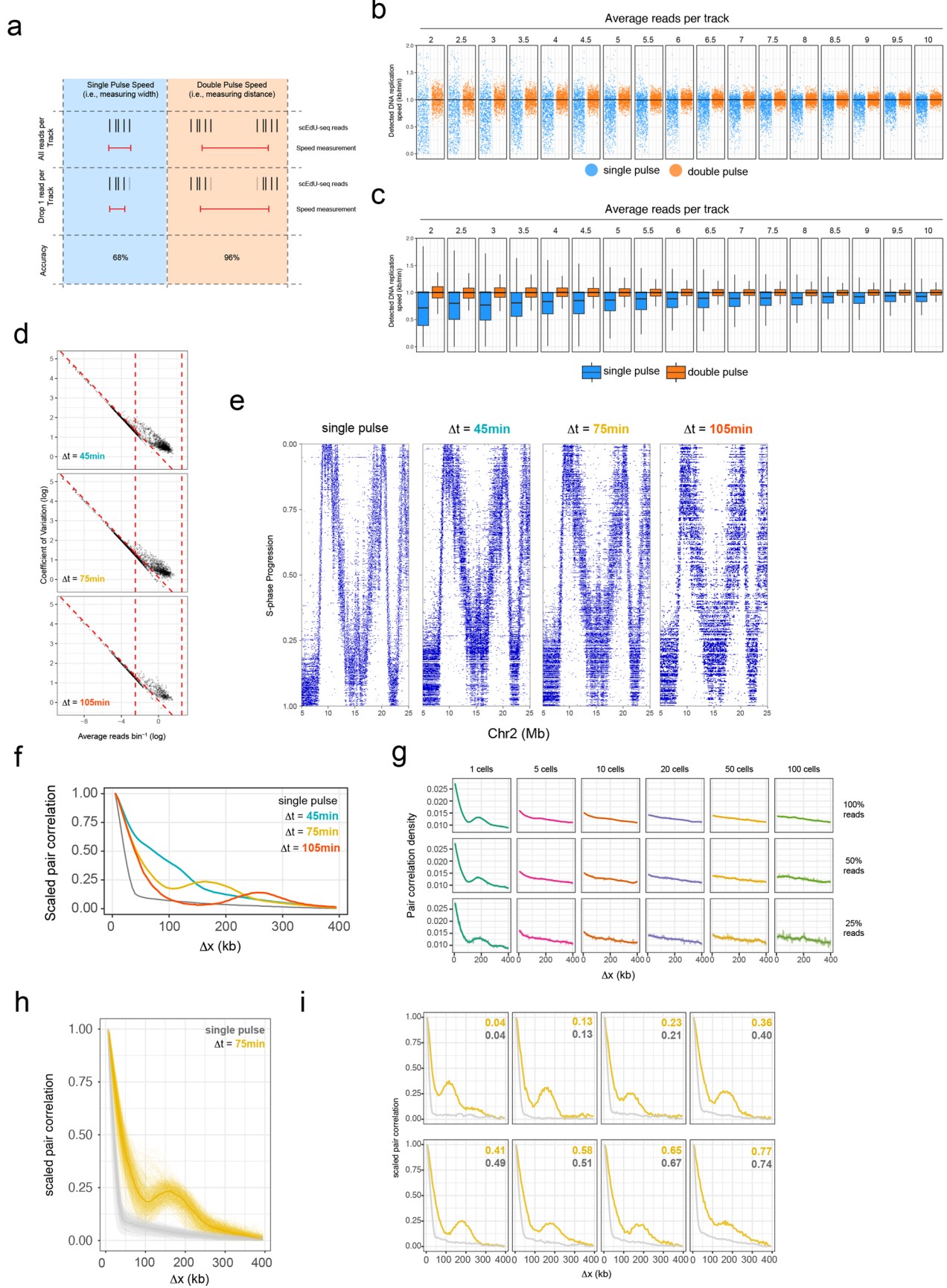

**Extended Data Fig. 4 | See next page for caption.**

**Extended Data Fig. 4 | Comparing of single-pulse to double-pulse labelling scheme. a**. Cartoon representation for DNA replication speed calculations using single and double pulse labeling and the effect of sampling on the accuracy on the detected speeds. **b**. Simulations of detected DNA replication speeds sampled 1000 tracks with the indicated average number of reads per condition (Each dot indicates a single replication track). **c**. Simulations of detected DNA replication speeds sampled 1000 tracks with the indicated average number of reads per conditions plotted as a boxplot (the box indicates IQR and median and the range is Tukey). **d**. Coefficient of Variation versus average reads per bin for all $\Delta t = 45$ min, $\Delta t = 75$ min and $\Delta t = 105$ min double pulse library scEdU-seq cells. Each dot is a single cell and the top area between three dashed lines contain selected cells for subsequent analysis. **e**. Heatmap of scEdU-seq maximum normalized log counts for all single pulse, $\Delta t = 45$ min, $\Delta t = 75$ min and $\Delta t = 105$ min double pulse cells ordered according to S-phase progression (*y-axis*)

and binned per 5 kb bins (x-axis) for 20 Megabase of chromosome 2. **f**. Overlay of averaged pair correlation for all single pulse, $\Delta t = 45$ min, $\Delta t = 75$ min and $\Delta t = 105$ min conditions. **g**. Pair correlation analysis of a double pulse $\Delta t = 75$ min EdU treated RPE-1 cells with read downsampling and *in silico* bulk conditions. From left to right, single cell versus in silico bulk (5, 10, 20, 50, 100 cells) of nearest neighbor cells in S-phase progression space. From top to bottom, downsampling of unique reads per condition. **h**. Line plots of the pair correlations of single pulse (grey), $\Delta t = 75$ min double labeling (yellow). Every thin line is a single RPE-1 cell where the x-axis shows the distance and the y-axis the range-scaled density. Bold lines (grey and yellow) indicate the mean density of each labeling condition. **i**. Representative single cell pair correlation traces for single pulse (15 min EdU) and double pulse labeling ($\Delta t = 75$ min). Each plot contains a matched cell from a 10 percentile S-phase Progression bin for both single and double pulse labels (grey and yellow numbers indicate S-phase progression for each single cell).

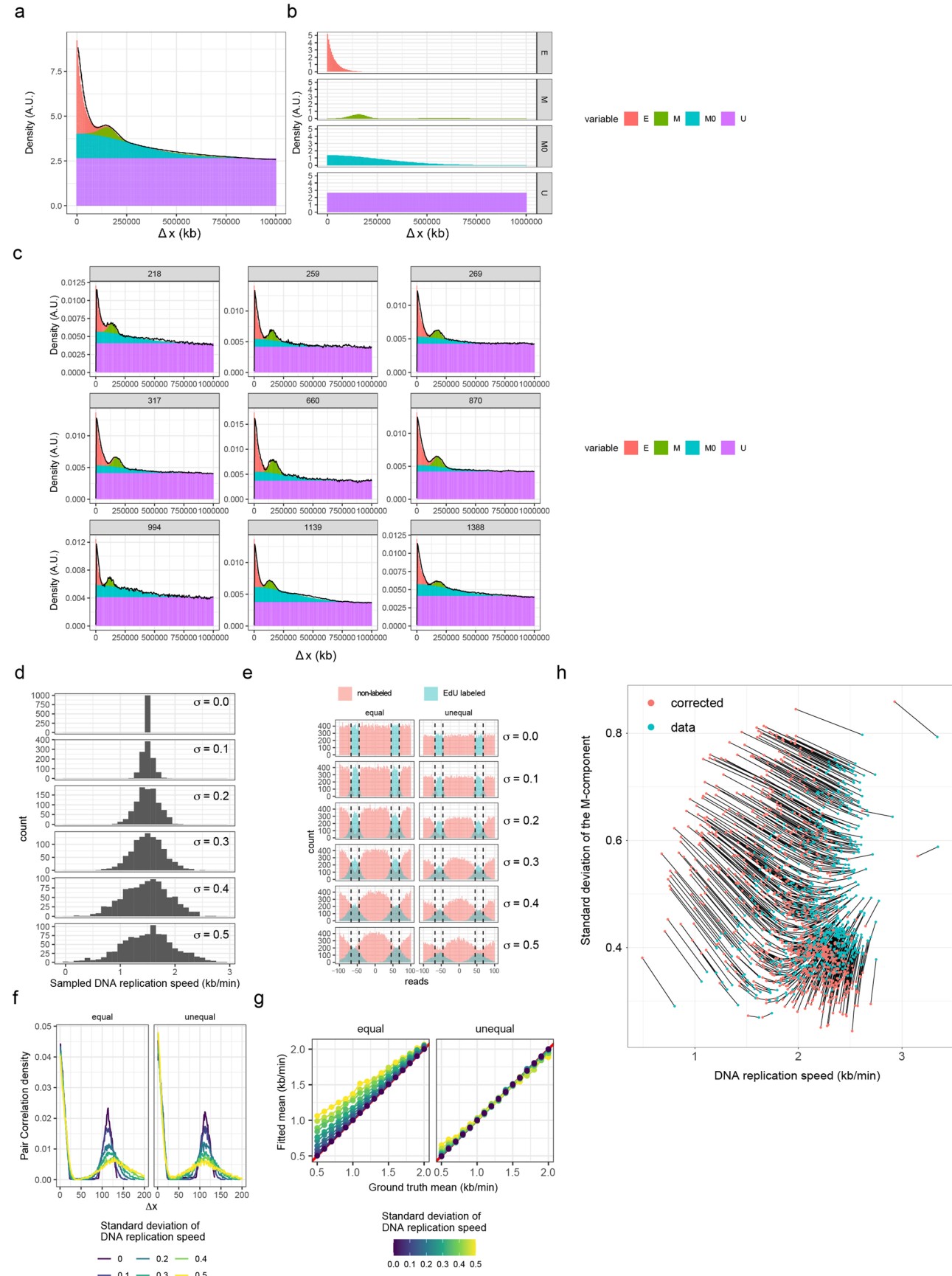

**Extended Data Fig. 5 | See next page for caption.**

**Extended Data Fig. 5 | Modelling and simulation of double pulse scEdU-seq data. a**. Four component mixture model fit for all aggregate $\Delta t = 75$ min double pulse RPE-1 cells. Aggregate values per component per binned distance in kilobase (E= Exponent, N=normal (speed), M0 = half-normal, U= Uniform components). **b**. Four component mixture model fit for all aggregate $\Delta t = 75$ min double pulse RPE-1 cells split between all components. (E= Exponent, N=normal (speed), M0 = half-normal, U= Uniform components). **c**. Representative single cells with Four component mixture model fit for $\Delta t = 75$ min double pulse RPE-1 cells. (E= Exponent, N=normal (speed), M0 = half-normal, U= Uniform components). **d**. Simulated DNA replication speeds drawn from a truncated normal distribution with a mean of 1.5 kb/min and increasing standard deviation (0-0.5 kb/min). **e**. Density of reads (y-axis) aggregated per standard deviation of the underlying speed distribution for the piece of simulated genome (x-axis), colored by EdU (inside window) or non-labeled (outside window) and faceted by the poisson intensity assumption. **f**. Line plots of the pair correlation for simulated double pulse labeling for equal and unequal sampling with increasing DNA replication speed variance. The x-axis shows the binned distance and the y-axis the range-scaled density. Line colors indicate the underlying DNA replication speed variance. **g**. Average of the between-pulse-distances from the simulated data (y-axis) versus the ground-truth simulated mean (x-axis). Line colors indicate the ground-truth simulated mean variance and the facets show the equal and unequal sampling assumptions. **h**. Standard deviation (y-axis) versus the mean (x-axis) of the speed component where every dot is colored indicating whether the fitted or corrected values are plotted, and where dots representing the same cell before and after correction are connected by a line.

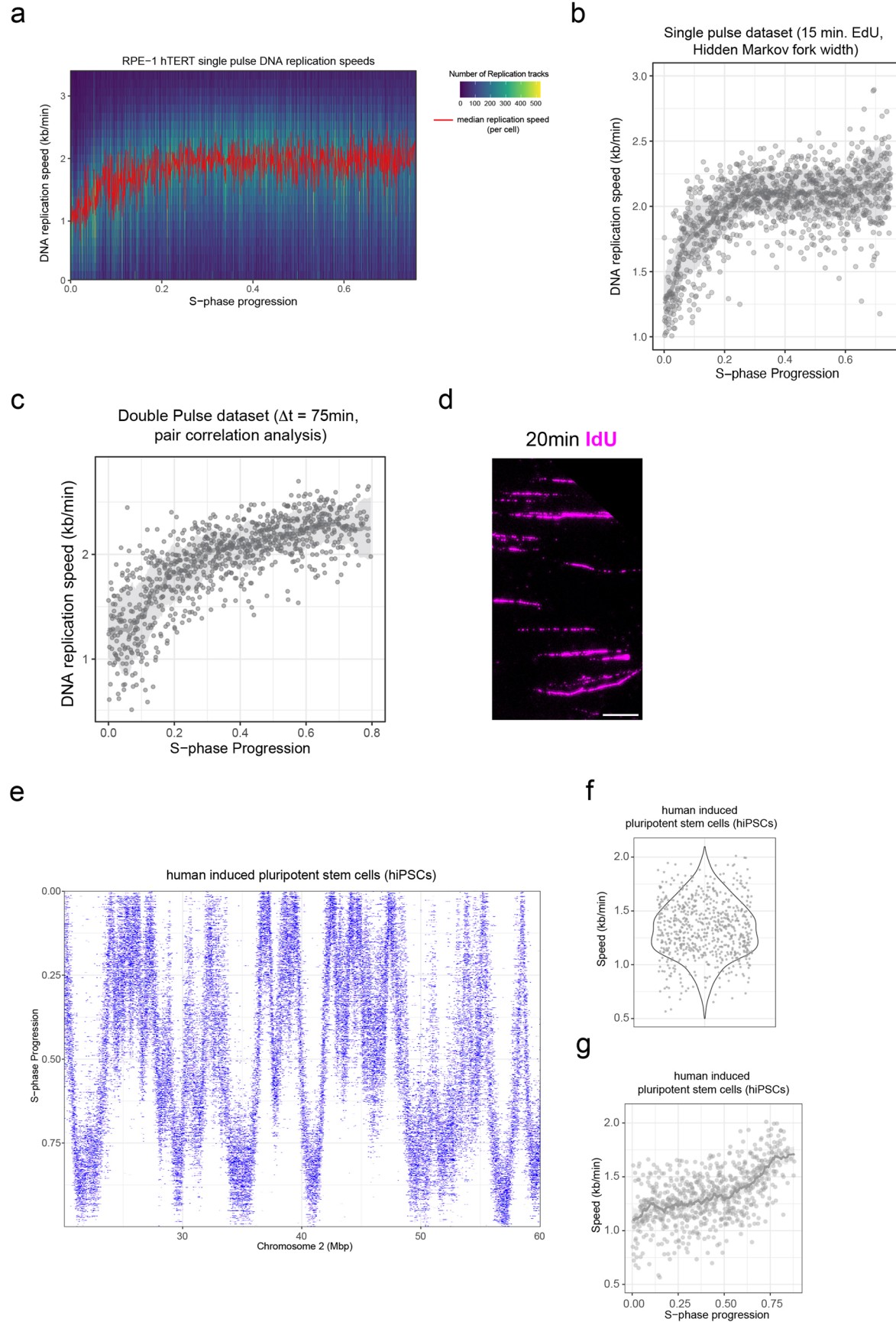

**Extended Data Fig. 6 | See next page for caption.**

**Extended Data Fig. 6 | Validation of DNA replication speed increase over S-phase. a**. RPE-1 cells treated with a 15 min EdU single pulse. Number of forks (color scale) that have a specific DNA replication speed (y-axis) versus S-phase progression (x-axis), red line indicates the median DNA replication speed per cell. **b**. DNA replication speed over S-phase in RPE-1 treated with DMSO (gray, n = 326) subjected to $\Delta t = 75$ min. labeling scheme and analyzed by pair correlation analysis. **c**. DNA replication speeds over S-phase in RPE-1 treated with DMSO (gray, n = 1350) subjected to single pulse (15 min EdU) and extracted as width from Hidden Markov Model. **d**. Representative Field of view of DNA combing from RPE-1 hTERT cells labeled with 20 minutes of IdU (250 μM) and counterstained with an IdU antibody (n = 3). **e**. Heatmap of scEdU-seq maximum normalized log counts for $\Delta t=75$ min human induced pluripotent stem cells (hiPSC) ordered according to S-phase progression (y-axis) and binned per 5 kb bins (x-axis) for 20 Megabase of chromosome 2. **f**. Violin plot depicting DNA replication speeds in hiPSC (every dot indicates a single cell). **g**. DNA replication speed over S-phase in hiPSCs treated with DMSO subjected to $\Delta t = 75$ min. labeling scheme. Speed estimates in kb/min (y-axis) over S-phase progression (x-axis). Every dot is a cell, the line indicates a rolling-window median smooth and the ribbon the standard deviation around the median.

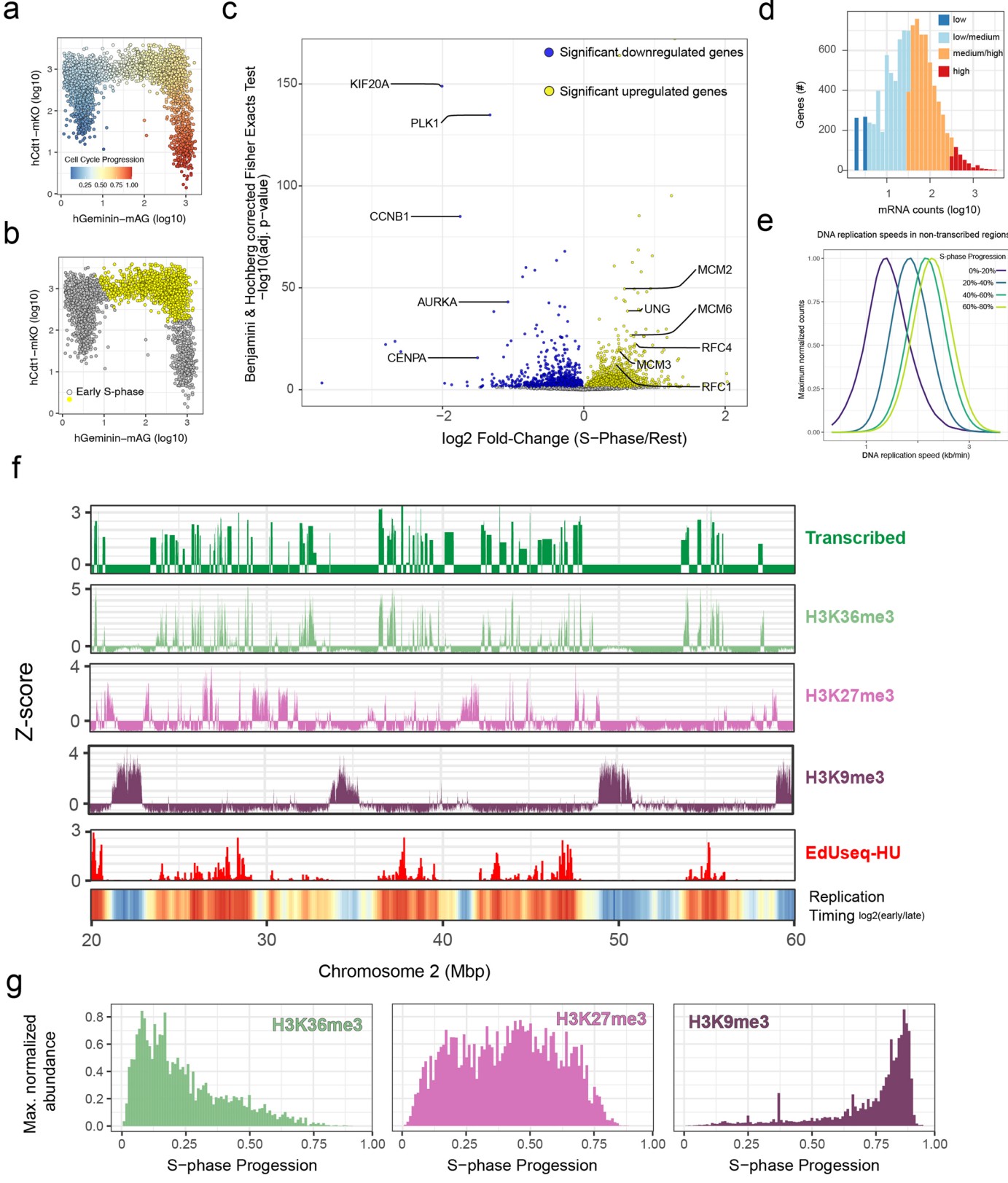

**Extended Data Fig. 7 | Effect of transcription and histone modifications on DNA replication speed. a.** FUCCI reporter intensities with pseudotime ordering (wanderlust) based on scEUseq counts (Battich et al., 2020). **b.** Early S-phase cells selected for gene expression analysis from scEUseq data in Fig. 3c. **c.** Differentially expressed genes between early S-phase cells versus the rest. Volcano plot displaying log2 Fold Change (x-axis) versus adjusted Fisher exact test (Benjamini-Hochberg correction for multiple testing). Each dot represents a gene and is colored by significance (adj. p-value < 0.05) and up/down sign. **d.** Distribution of log10-transformed nascent mRNA counts for pseudo-bulked S-phase cells **e.** Maximum normalized distribution of DNA replication speeds in the indicated bins of S-phase Progression of RPE-1 hTERT cells in non-transcribed (grey point and line) and transcribed regions of the genome. **f.** Z-scored tracks for transcribed, H3K36me3, H3k27me3, H3K9me3 regions and Replication Timing (log2(early/late) Repli-Seq) track for RPE-1 hTERT cells displaying 50 Mb of chromosome 2. **g.** Relative chromatin mark abundance (y-axis) along S-phase progression (x-axis) split by H3K36me3, H3K9me3 and H3K27me3.

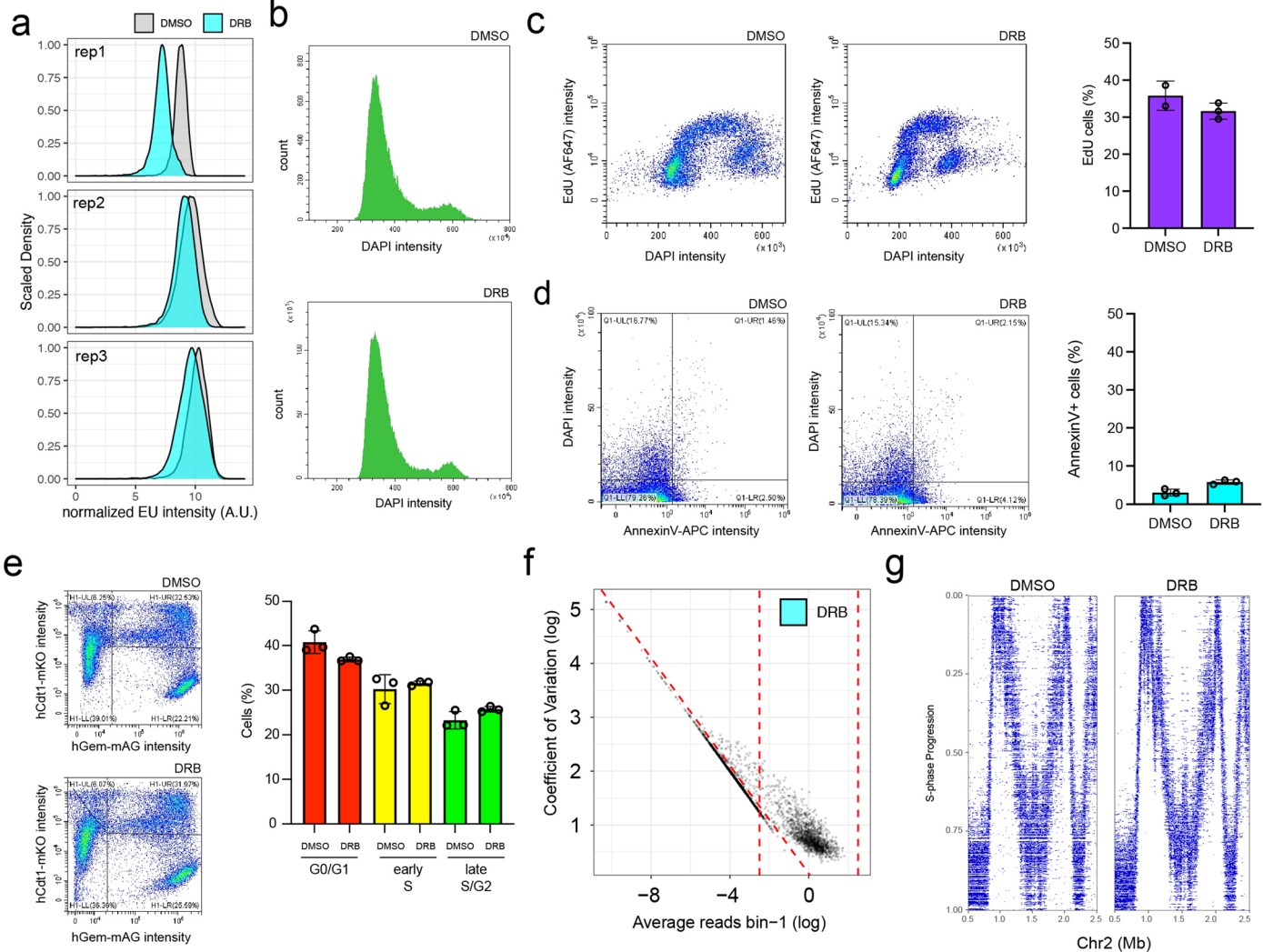

**Extended Data Fig. 8 | Inhibition of transcription using DRB. a**. Nascent RNA labeling with EU (30 min) on fixed RPE-1 cells treated with DMSO or DRB (1 hr). **b**. Representative DNA content (DAPI) intensities for fixed RPE-1 cells treated with DMSO or DRB (1 hr). **c**. Representative EdU (30 min) intensities for fixed RPE-1 cells treated with DMSO or DRB for 1 hr (*left*). Quantification of EdU[+] cells, each dot indicates a biological replicate (*right*) (n = 3, mean +/- SD). **d**. Representative AnnexinV/DAPI intensities on living RPE-1 cells treated with DMSO or DRB for 1 hr (*left*). Quantification of AnnexinV/DAPI[+] cells, each dot indicates a biological replicate (*right*) (n = 3, mean +/- SD). **e**. Representative FUCCI reporter intensities on fixed RPE-1 cells treated with DMSO or DRB (*left*). Quantification of G0/G1, early S and late S-phase/G2 cells, each dot indicates a biological replicate (*right*) (n = 3, mean +/- SD). **f**. Coefficient of Variation (y-axis) versus average reads per bin (x-axis) for all DRB-treated $\Delta t$ = 75 min scEdU-seq cells. Each dot is a single cell and the top area between three dashed lines contain selected cells for subsequent analysis. **g**. Heatmap of scEdU-seq maximum normalized log counts for DMSO and DRB-treated $\Delta t$ = 75 min double pulse cells ordered according to S-phase progression (*y-axis*) and binned per 5 kb bins (x-axis) for 2 megabase of chromosome 2.

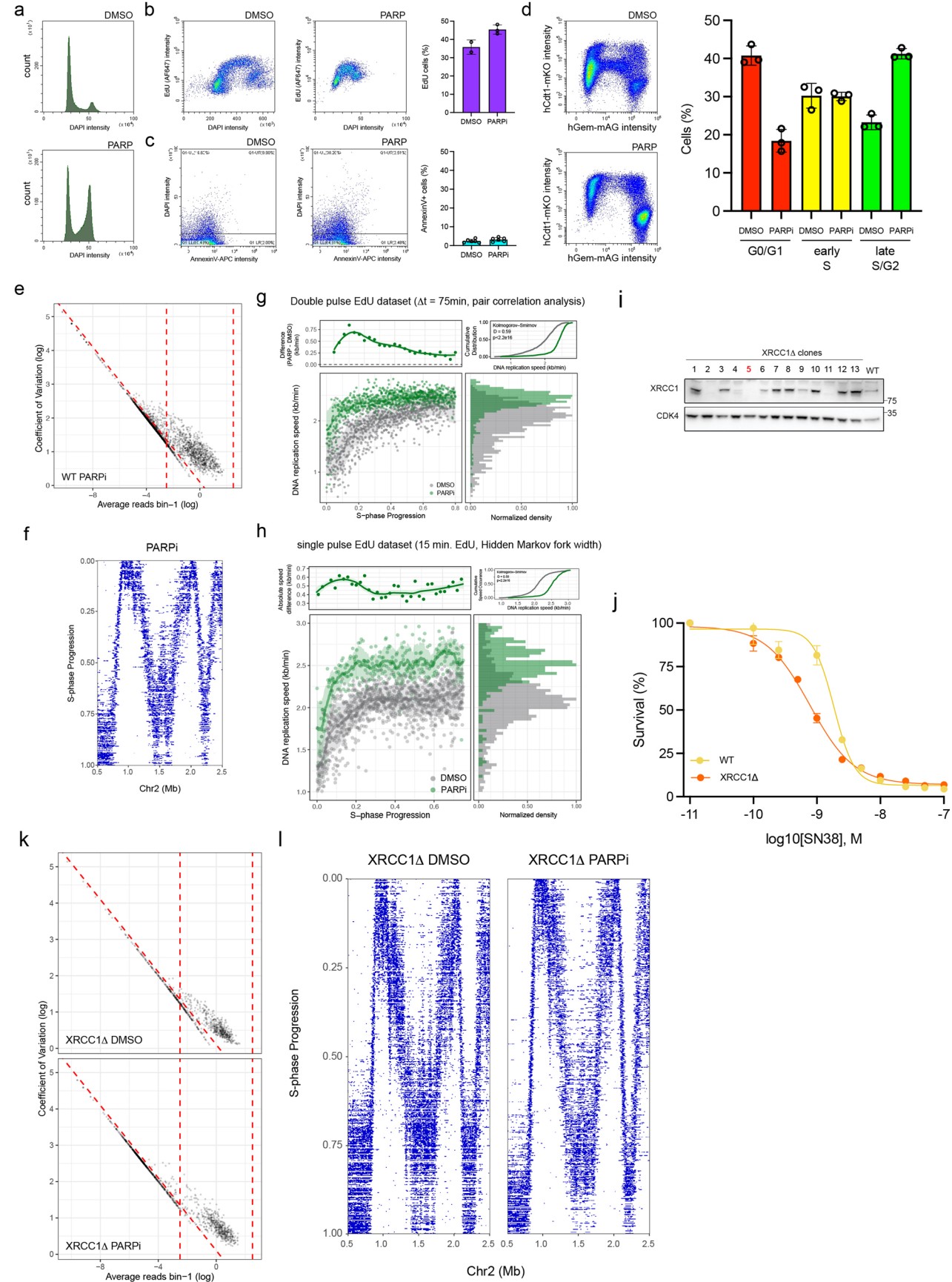

**Extended Data Fig. 9 | See next page for caption.**

**Extended Data Fig. 9 | Perturbing DNA damage sensing. a**. Representative DNA content (DAPI) intensities for fixed RPE-1 cells treated with DMSO or 24 hr PARPi. **b**. Representative EdU (30 min) intensities for fixed RPE-1 cells treated with DMSO or 24 hr PARPi (*left*). Quantification of EdU⁺ cells, each dot indicates a biological replicate (*right*) (n = 3, mean +/- SD). **c**. Representative AnnexinV/DAPI intensities on living RPE-1 cells treated with DMSO or 24 hr PARPi (*left*). Quantification of AnnexinV/DAPI⁺ cells, each dot indicates a biological replicate (*right*) (n = 3, mean +/- SD). **d**. Representative FUCCI reporter intensities on fixed RPE-1 cells treated with DMSO or 24 hr PARPi (*left*). Quantification of G0/G1, early S and late S-phase/G2 cells, each dot indicates a biological replicate (*right*) (n = 3, mean +/- SD). **e**. Coefficient of Variation (y-axis) versus average reads per bin (x-axis) for all 24 hr PARP inhibitor treated $\Delta t$ = 75 min scEdU-seq cells. Each dot is a single cell and the top area between three dashed lines contain selected cells for subsequent analysis. **f**. Heatmap of scEdU-seq maximum normalized log counts for PARP-treated $\Delta t$ = 75 min double pulse cells ordered according to S-phase progression (*y-axis*) and binned per 5 kb bins (x-axis) for 2 megabase of chromosome 2. **g**. The data from Fig. 4b. Generated by double pulse EdU and pair correlation analysis. DNA replication speed over S-phase in RPE-1 treated with DMSO (gray, n = 326) or 24 hr PARPi (green, n = 766) subjected to $\Delta t$ = 75 min. labeling scheme (bottom-left). Difference in DNA replication speeds between DMSO and PARP in kb/min (y-axis) over S-phase progression (x-axis, top-left), marginal density

(x-axis) of DNA replication speeds in kb/min (y-axis) colored for DMSO- (gray) or PARP-treated cells (green, bottom-right) and cumulative distribution of marginal speed density (top-right). **h. T**he data generated by single pulse EdU and analysis of forks width by Hidden Markov Model. DNA replication speeds over S-phase in RPE-1 treated with DMSO (gray, n = 1350) or 24 hr PARPi (green, n = 339) subjected to single pulse (15 min EdU). Difference in DNA replication speeds between DMSO and PARP in kb/min (y-axis) over S-phase progression (x-axis, top-left), marginal density (x-axis) of DNA replication speeds in kb/min (y-axis) colored for DMSO- (gray) or PARP-treated cells (green, bottom-right) and cumulative distribution of marginal speed density (top-right). **i**. Western Blot analysis of XRCC1 RPE-1 clones with XRCC1 antibody to validate knockout status and CDK4 as a loading control (n = 1). **j**. Viability of XRCC1 RPE-1 and parental RPE-1 cells in response to increasing concentrations of Topoisomerase I poison, SN-38 (n = 3). **k**. Coefficient of Variation (y-axis) versus average reads per bin (x-axis) for all XRCC1 RPE-1 DMSO and 4 hr PARPi-treated t = 75 min scEdU-seq cells. Each dot is a single cell and the top area between three dashed lines contain selected cells for subsequent analysis. **l**. Heatmap of scEdU-seq maximum normalized log counts for XRCC1 RPE-1 DMSO and PARPi-treated $\Delta t$ = 75 min scEdU-seq cells ordered according to S-phase progression (*y-axis*) and binned per 5 kb bins (x-axis) for 2 megabases of chromosome 2.

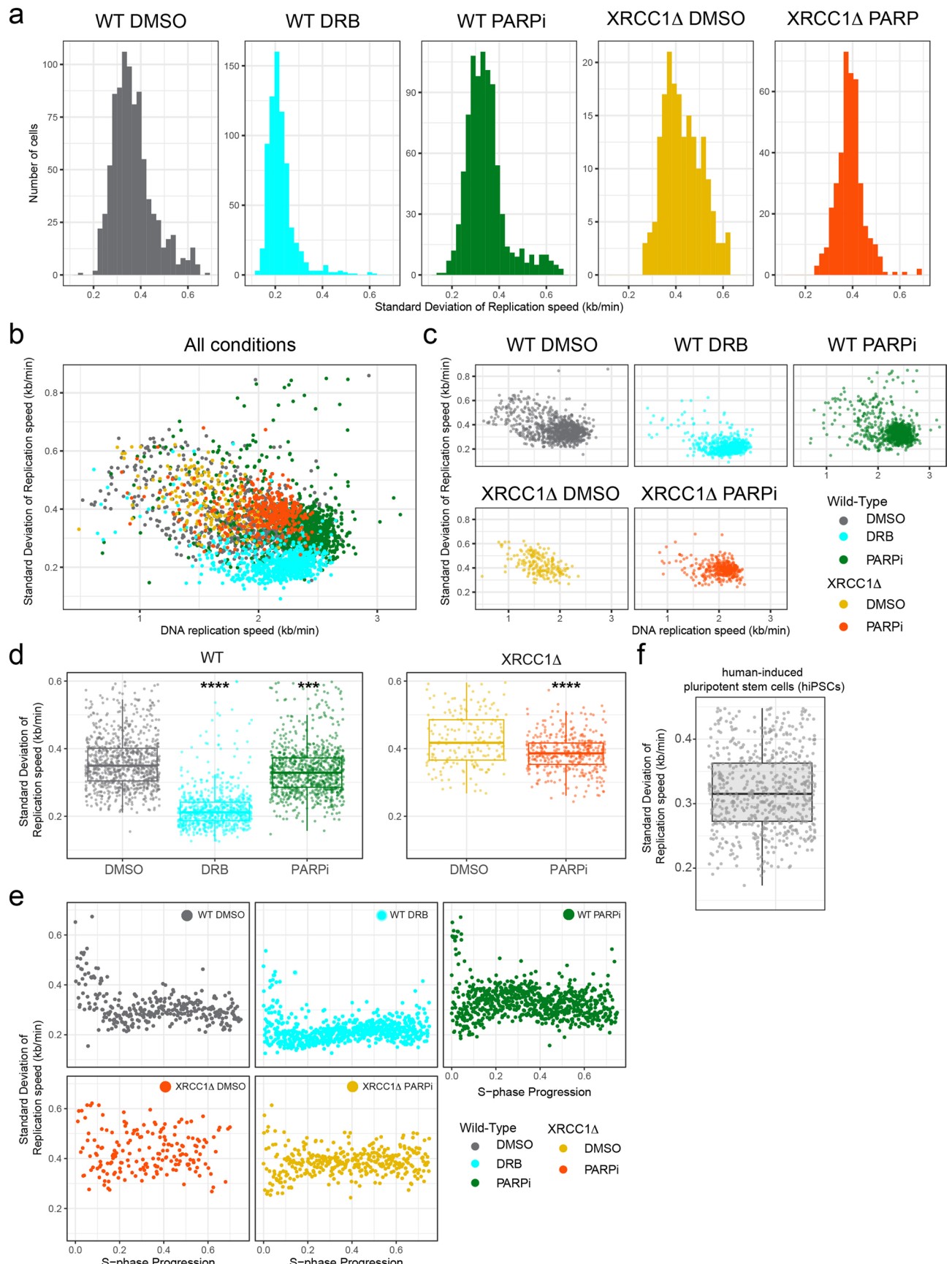

**Extended Data Fig. 10 | See next page for caption.**

**Extended Data Fig. 10 | Variability of DNA replication speeds within single cell. a**. Distribution replication speed variance (kb/min) from single cells over all tested conditions in this work. **b**. Scatter plot with DNA replication speed (kb/min, x-axis) and DNA replication speed variance (kb/min, y-axis) from single cells containing all conditions tested in this work. **c**. Scatter plot with DNA replication speed (kb/min, x-axis) and DNA replication speed variance (kb/min, y-axis) from single cells split by treatment and genotype. **d**. Variability of DNA replication speeds within single cells (kb/min, y-axis) split over Wild-type and XRCC1Δ in RPE-1 cells labeled with Δt= 75 min. EdU scheme. e. Variability of DNA replication speeds within single cells (kb/min, y-axis) over S-phase Progression (x-axis) split over wild-type DMSO (n-=326), wild-type (n = 713 DRB,wild-type PARP (n = 766) i, XRCC1Δ DMSO (n = 187) and XRCC1Δ PARPi (n = 393) conditions in RPE-1 cells

labeled with Δt = 75 min. EdU scheme. Two sided Student T-test with multiple testing correction was performed. The box of the boxplot is defined by the median ± IQR and whiskers are 1.5X IQR. In wild-type cells, Two-sided Students T-test (Bonferroni-corrected) yield pvalue < 1.31e- 4 (DMSO vs PARP) and pvalue < 1.90e-225 (DMSO vs DRB) For XRCC1Δ RPE-1 cells, Two-sided Students T-test (Bonferroni-corrected) yielded a pvalue < 3.60e-8 (DMSO vs PARP) **e**. Variability of DNA replication speeds within single cells (kb/min, y-axis) over S-phase (x-axis))split over Wild-type and XRCC1Δ in RPE-1 cells labeled with Δt= 75 min. EdU scheme. **f**. Variability of DNA replication speeds within single hiPSCs (kb/min, y-axis) in hiPSCs labeled with Δt= 75 min. EdU scheme. The box of the boxplot is defined by the median ± IQR and whiskers are 1.5X IQR.

# Reporting Summary

## Statistics

For all statistical analyses, confirm that the following items are present in the figure legend, table legend, main text, or Methods section.

| n/a | Confirmed | |
|---|---|---|
| ☐ | ☒ | The exact sample size (*n*) for each experimental group/condition, given as a discrete number and unit of measurement |
| ☐ | ☒ | A statement on whether measurements were taken from distinct samples or whether the same sample was measured repeatedly |
| ☐ | ☒ | The statistical test(s) used AND whether they are one- or two-sided<br>*Only common tests should be described solely by name; describe more complex techniques in the Methods section.* |
| ☐ | ☒ | A description of all covariates tested |
| ☐ | ☒ | A description of any assumptions or corrections, such as tests of normality and adjustment for multiple comparisons |
| ☐ | ☒ | A full description of the statistical parameters including central tendency (e.g. means) or other basic estimates (e.g. regression coefficient) AND variation (e.g. standard deviation) or associated estimates of uncertainty (e.g. confidence intervals) |
| ☐ | ☒ | For null hypothesis testing, the test statistic (e.g. *F*, *t*, *r*) with confidence intervals, effect sizes, degrees of freedom and *P* value noted<br>*Give P values as exact values whenever suitable.* |
| ☐ | ☒ | For Bayesian analysis, information on the choice of priors and Markov chain Monte Carlo settings |
| ☒ | ☐ | For hierarchical and complex designs, identification of the appropriate level for tests and full reporting of outcomes |
| ☒ | ☐ | Estimates of effect sizes (e.g. Cohen's *d*, Pearson's *r*), indicating how they were calculated |

*Our web collection on statistics for biologists contains articles on many of the points above.*

## Software and code

Policy information about availability of computer code

| Data collection | Sequencing data were collected on the Illumina NextSeq 500 or 2000 (NextSeq Control Software version 2.2.0.4), using standard software for basecalling (RTA version 2.4.11). Sample demultiplexing was performed using bcl2fastq (v2.20.0.422).<br><br>FACS data were collected on a BD Influx (BD FACS Sortware version 1.2.0.142) for sorting experiments<br><br>FACS data were collected on a Beckman-Coulter Cytoflex (CytExpert software version 2.5, March 2022) for flow cytometry experiments<br><br>Western Blot development was performed on a Bio-Rad Chemi-Doc MP with manufacturers software |
|---|---|
| Data analysis | Data were analyzed using a combination of publicly available and custom software.<br>Publicly available software included: SingleCellMultiOmics (v0.1.2.5), cutadapt (version 3.2), bwa (version 2.7.6a), python (versions 3.7.3 & 3.8.2), round_0.20-0, umap_0.2.8.0, mhsmm_0.4.16, mvtnorm_1.1-3, flexmix_2.3-17, lattice_0.20-45, Rcpp_1.0.7, forcats_0.5.1, stringr_1.4.0, dplyr_1.0.9, purrr_0.3.4, readr_2.1.2, tidyr_1.2.0, tibble_3.1.7, ggplot2_3.3.6, tidyverse_1.3.1, data.table_1.14.2, R version 4.2.0 (2022-04-22)<br>All custom scripts to process raw data and generate figures are available at https://github.com/vincentvbatenburg/scEdU-seq |

For manuscripts utilizing custom algorithms or software that are central to the research but not yet described in published literature, software must be made available to editors and reviewers. We strongly encourage code deposition in a community repository (e.g. GitHub). See the Nature Portfolio guidelines for submitting code & software for further information.

## Data

Policy information about availability of data

All manuscripts must include a data availability statement. This statement should provide the following information, where applicable:
- Accession codes, unique identifiers, or web links for publicly available datasets
- A description of any restrictions on data availability
- For clinical datasets or third party data, please ensure that the statement adheres to our policy

Raw sequencing data, metadata and count tables have been made available in the Gene Expression Omnibus under the accession number GSE211037.

Data for comparisons to scEUseq and scRepli-Seq were downloaded from Gene Expression Omnibus accessions GSE128365 and GSE108556. Raw sequencing data of DNA replication origins was downloaded from SRA (PRJNA397123). Data for Replication timing was downloaded from the 4D nucleome project (4DNBSKYMY5XL)

## Human research participants

Policy information about studies involving human research participants and Sex and Gender in Research.

| | |
|---|---|
| Reporting on sex and gender | n/a |
| Population characteristics | n/a |
| Recruitment | n/a |
| Ethics oversight | n/a |

Note that full information on the approval of the study protocol must also be provided in the manuscript.

# Field-specific reporting

Please select the one below that is the best fit for your research. If you are not sure, read the appropriate sections before making your selection.

☒ Life sciences          ☐ Behavioural & social sciences          ☐ Ecological, evolutionary & environmental sciences

For a reference copy of the document with all sections, see nature.com/documents/nr-reporting-summary-flat.pdf

# Life sciences study design

All studies must disclose on these points even when the disclosure is negative.

| | |
|---|---|
| Sample size | No statistical methods were used to predetermine sample size. The number of cells analyzed was chosen to enable sufficient technical validation and benchmarking of scEdUseq. A minimum of 100 cells per replicate was used to enable proper ordering of cells along S-phase. |
| Data exclusions | All raw data are uploaded in public repositories. In downstream analyses, cells were excluded if they did not meet two quality control thresholds: a minimum number of average reads per bin per cell and a coefficient of variation clearly deviating from Poisson noise. These thresholds removed cells where the library construction and/or EdU incorporation failed. The values of these cutoffs were slightly adjusted for each plate and sample type to compensate for differences in sequencing depth and EdU incorporation. |
| Replication | For the hTERT-RPE-1 cell-cycle experiments, the DMSO and PARP treated cells were measured in three independent experiments, and the DRB and XRCC1 experiments were measured in one experiment. All attempts at replication were successful. |
| Randomization | No randomization was performed. Randomization was not necessary as all samples were processed in an automarted fashion by robotic liquid handlers. |
| Blinding | No blinding was performed since we performed unsupervised analysis techniques (e.g., clustering and dimensionality reduction) |

# Reporting for specific materials, systems and methods

We require information from authors about some types of materials, experimental systems and methods used in many studies. Here, indicate whether each material, system or method listed is relevant to your study. If you are not sure if a list item applies to your research, read the appropriate section before selecting a response.

## Materials & experimental systems

| n/a | Involved in the study |
|---|---|
| ☐ | ☒ Antibodies |
| ☐ | ☒ Eukaryotic cell lines |
| ☒ | ☐ Palaeontology and archaeology |
| ☒ | ☐ Animals and other organisms |
| ☒ | ☐ Clinical data |
| ☒ | ☐ Dual use research of concern |

## Methods

| n/a | Involved in the study |
|---|---|
| ☒ | ☐ ChIP-seq |
| ☐ | ☒ Flow cytometry |
| ☒ | ☐ MRI-based neuroimaging |

## Antibodies

| Antibodies used | XRCC1 (abcam, ab1838) (1:1000)<br>CDK4 (Santa Cruz Biotechnologies, sc-260) (1:1000)<br>pan ADP-ribose (MABE1016, Sigma-Aldrich) (1:1500)<br>gH2AX (Millipore, JBW301) (1;!000)<br>anti-H3K9Me3 (Abcam, ab8898, 1:100),<br>anti-H3K27Me3 (Cell Signaling Technologies, C36B11, 1:200) and<br>anti-H3K36Me3 (Thermo Fisher, MA5-24687, 1:2000). |
|---|---|
| Validation | XRCC1 (https://www.abcam.com/xrcc1-antibody-33-2-5-ab1838.html), Hoch et al., Nature 2017. CDK4 (https://www.scbt.com/p/cdk4-antibody-c-22), van den berg et al., NAR, 2016. pan ADP-ribose (https://www.merckmillipore.com/NL/en/product/Anti-pan-ADP-ribose-binding-reagent,MM_NF-MABE1016) Hoch et al., Nature 2017. gH2AX (https://www.merckmillipore.com/NL/en/product/Anti-phospho-Histone-H2A.X-Ser139-Antibody-clone JBW301,MM_NF-05-636) van den Berg et al., NAR, 2016 anti-H3K9Me3 (Abcam, ab8898, 1:100),  Zeller et al., Nature Genetics 2022, anti-H3K27Me3 (Cell Signaling Technologies, C36B11, 1:200) Zeller et al., Nature Genetics 2022, and anti-H3K36Me3 (Thermo Fisher, MA5-24687, 1:2000) Zeller et al., Nature Genetics 2022. |

## Eukaryotic cell lines

Policy information about cell lines and Sex and Gender in Research

| Cell line source(s) | hTERT RPE-1 FUCCI and hTERT RPE-1 iCut cells were both obtained from the Medema lab at the Netherlands Cancer Institute. |
|---|---|
| Authentication | None of the cell lines were authenticated. |
| Mycoplasma contamination | All cell lines routinely tested negative for Mycoplasma contamination |
| Commonly misidentified lines<br>(See ICLAC register) | No commonly misidentified cell lines were used in the study. |

## Flow Cytometry

### Plots

Confirm that:

☒ The axis labels state the marker and fluorochrome used (e.g. CD4-FITC).

☒ The axis scales are clearly visible. Include numbers along axes only for bottom left plot of group (a 'group' is an analysis of identical markers).

☒ All plots are contour plots with outliers or pseudocolor plots.

☒ A numerical value for number of cells or percentage (with statistics) is provided.

### Methodology

| Sample preparation | hTERT RPE-1 FUCCI cells: Cells were dissociated to a single-cell suspension using TrypLE, washed and resuspended in PBS containing DAPI as a viability stain and BSA to reduce aggregation. Cells were passed through a 20-micron mesh before sorting. |
|---|---|
| Instrument | BD Influx and Beckman-Coulter Cytoflex |
| Software | BD FACS Sortware 1.2.0.142 and  CytExpert software |
| Cell population abundance | Cell population purity and abundance was not explicitly determined after sorting. FACS was primarily used to i) distribute single cells into individual wells of a 384-well plate for subsequent processing, ii) measure the fluorescence of cell-cycle progression markers of these cells,. |
| Gating strategy | hTERT RPE-1 Doublets, debris, and dead cells were excluded by gating forward and side scatter in combination with |

Gating strategy | the DAPI channel.  Example gating strategy for is  provided in Extended Data Figures 1a and 1c.

☒ Tick this box to confirm that a figure exemplifying the gating strategy is provided in the Supplementary Information.

