## [Peer Review File · Nature Methods]

Peer Review Information

Manuscript Title: Quantifying DNA replication speeds in single cells by scEdU-seq

Corresponding author name(s): Alexander van Oudenaarden

Editorial Notes:

Reviewer Comments & Decisions:

Decision Letter, initial version:

16th Nov 2023

Dear Dr van Oudenaarden,

Your Article, "Quantifying DNA replication speeds in single cells by scEdU-seq", has now been seen by 3 reviewers. As you will see from their comments below, although the reviewers find your work of considerable potential interest, they have raised a number of technical concerns. We are interested in the possibility of publishing your paper in Nature Methods, but would like to consider your response to these concerns before we reach a final decision on publication.

We therefore invite you to revise your manuscript to address these concerns. We are committed to providing a fair and constructive peer-review process. Do not hesitate to contact us if there are specific requests from the reviewers that you believe are technically impossible or unlikely to yield a meaningful outcome.

- * include a point-by-point response to the reviewers and to any editorial suggestions
- * please underline/highlight any additions to the text or areas with other significant changes to facilitate review of the revised manuscript
- * address the points listed described below to conform to our open science requirements

* ensure it complies with our general format requirements as set out in our guide to authors at www.nature.com/naturemethods

* resubmit all the necessary files electronically by using the link below to access your home page

[REDACTED]

We hope to receive your revised paper within 12 weeks. If you cannot send it within this time, please let us know. In this event, we will still be happy to reconsider your paper at a later date so long as nothing similar has been accepted for publication at Nature Methods or published elsewhere.

OPEN SCIENCE REQUIREMENTS

REPORTING SUMMARY AND EDITORIAL POLICY CHECKLISTS

IMAGE INTEGRITY

When submitting the revised version of your manuscript, please pay close attention to our Digital Image Integrity Guidelines and to the following points below:

-- that unprocessed scans are clearly labelled and match the gels and western blots presented in

figures.

- that control panels for gels and western blots are appropriately described as loading on sample processing controls
- all images in the paper are checked for duplication of panels and for splicing of gel lanes.

DATA AVAILABILITY

All novel DNA and RNA sequencing data, protein sequences, genetic polymorphisms, linked genotype and phenotype data, gene expression data, macromolecular structures, and proteomics data must be deposited in a publicly accessible database, and accession codes and associated hyperlinks must be provided in the "Data Availability" section.

CODE AVAILABILITY

Please include a "Code Availability" subsection in the Online Methods which details how your custom

code is made available. Only in rare cases (where code is not central to the main conclusions of the paper) is the statement "available upon request" allowed (and reasons should be specified).

For more information on our code sharing policy and requirements, please see:
<https://www.nature.com/nature-research/editorial-policies/reporting-standards#availability-of-computer-code>

SUPPLEMENTARY PROTOCOL

To help facilitate reproducibility and uptake of your method, we ask you to prepare a step-by-step Supplementary Protocol for the method described in this paper. We encourage authors to share their step-by-step experimental protocols on a protocol sharing platform of their choice and report the protocol DOI in the reference list. Nature Portfolio 's Protocol Exchange is a free-to-use and open resource for protocols; protocols deposited in Protocol Exchange are citable and can be linked from the published article. More details can found at www.nature.com/protocolexchange/about.

ORCID

Nature Methods is committed to improving transparency in authorship. As part of our efforts in this direction, we are now requesting that all authors identified as 'corresponding author' on published papers create and link their Open Researcher and Contributor Identifier (ORCID) with their account on the Manuscript Tracking System (MTS), prior to acceptance. This applies to primary research papers only. ORCID helps the scientific community achieve unambiguous attribution of all scholarly contributions. You can create and link your ORCID from the home page of the MTS by clicking on 'Modify my Springer Nature account'. For more information please visit please visit www.springernature.com/orcid.

Sincerely,
Lei

Lei Tang, Ph.D.
Senior Editor
Nature Methods

Reviewers' Comments:

Reviewer #1:

Remarks to the Author:

van den Berg et al. present a clever and powerful approach for analyzing replication kinetic using metabolic labeling and deep sequencing. By combining standard approaches for pulse-labeling and isolating newly replicated DNA with EdU, single-cell characterization and isolation by FACS, and genome amplification, library creation and amplification, they design a pipeline that identifies the location of replication forks in single cells. The particular power of this approach is that it allows fork rate to be inferred from the distances between signals recorded during two sequential EdU pulses. This approach to measuring fork rate is a significant advance in the field because it allows the assignment of speeds to forks at known locations on a genome-wide scale. The method is used to investigate the affect of transcription of replication fork kinetics and will be useful for many other studies in the future. I have no substantive concerns.

Reviewer #2:

Remarks to the Author:

This manuscript describes a new method for measuring DNA replication and replication fork speed in single cells. The authors introduce both a single and double EdU pulse approach for this purpose. They investigate variability in replication speed along S phase and link it to transcription and DNA damage. This is a very useful and powerful technique, the paper reports novel and interesting results, it is well written and the analyses and presentation are of very high quality. I have a few questions/suggestions.

Figure 1b. Unclear why the authors chose to separate the data to early and late replication. In Repli-seq, early and late S phase fractions are collected however they are analyzed by comparing one to another rather than individually.

In Extended data figure 1b, the authors show lower correlation with Repli-seq for late S phase. Can they comment on that?

Extended data figure 1e and legend- this is not entirely clear. Can the authors elaborate on what the left and right plots are showing?

More generally, can the authors comment about how the choice of sorting gates influences their results?

Based on Figure 1f (and other figures), the data quality appears excellent, and not just for replication speed but for replication timing in general. The paper may benefit from emphasizing this aspect of the technique as well, including a quantitative comparison to other scRepli-seq approaches. That being said, based on figure 1f there does seem to be some loss of resolution at the very beginning and very end of S phase; this is expected, but can the authors comment on that?

In the last paragraph of the first results section, the authors talk about resolution of the technique. However, the results they present there pertain to downsampling of the data. I'm not sure if it's appropriate to refer to this analysis as "resolution". Perhaps the authors can be more specific in defining what they are analyzing here.

Overall, It seems to me that the authors developed a very powerful technique even just for measuring replication timing in single cells, but do not fully sieze on the opportunity to showcase its power.

The double pulse approach is clever and novel, and the correlation of replication speed with

transcription levels is very interesting. It's important to note that there have been previous studies that provided evidence for accelerated replication along S phase, for instance several studies described this in the context of the inactive X chromosome. While the current study certainly improves on those earlier studies with greater resolution and at the single-cell level, it's interesting to note that it validates and generalizes those earlier predictions. In this regard, did the authors see any unusual patterns of replication speeds on the X chromosome? Analyzing the active and inactive X chromosome separately could be an interesting avenue for a follow-up study.

The experiments connecting replication speeds to DNA damage are very interesting. One important question is whether transcription by itself slows replication or only in the context of DNA damage. It's unclear whether the authors have answered that question fully. They suggest that DNA damage is the mechanism causing slower replication at transcribed regions, however DNA damage occurs stochastically within transcribed regions, and some forms of DNA damage (and/or repair) are more common in late compared to early S phase or in heterochromatin compared to euchromatin. It becomes a rather complex system and the author's conclusions may be somewhat of an oversimplification at the moment. One way to further understand this would be to study transcription under the DNA damage conditions the authors used (including in single cells). An additional analysis could be of the stochasticity of replication speed influences by DNA damage; the authors mention variability in the last results paragraph but it seems this section could be made clearer and/or elaborated upon.

A limitation of the study is its being confined to a single cell line (RPE-1).

Reviewer #3:

Remarks to the Author:

van den Berg et al. have developed the scEdU-seq technology that, for the first time, allows to study replication fork progression in the context of the genome and interrogates about the cell-to-cell variability of this process in human cells. This technology is easily accessible to any laboratory and represents a significant advancement in the fields of DNA replication and repair as well as being relevant to any other field studying biological processes linked to cell division. The results presented in this study are intriguing and demonstrate how replication fork speed is increasing during the S phase and showed that the delay in early S can be, in part, due to the repair of DNA damage associated to the transcriptional process. However, in this manuscript, I would appreciate a better comparison with similar long-read sequencing technologies. Moreover, considering that the primary advantage of the technique is to obtain genomic information and cell to cell variability of the replication fork progression, a more detailed analysis of the resolution to determine replication fork speed at specific loci or genome features should be addressed. Finally, the authors should provide additional methodology details in the results section to facilitate the reading.

Main comments:

Figure 1A. When describing the methodology, the authors should indicate that they used cell sorting before performing single cell reactions. As it is written is not clear for the reader.

Results from Figure 2E and F, showed that replication fork speed values are higher when the time between pulses is longer. I wonder whether the replication fork speed can be overestimated when the

time between pulses is long because the longer the pulses are separated, the higher is the chance that two forks converge and are considered as one single fork instead of two. Could this bias be corrected by filtering out forks in which the width of any of the two EdU pulses is longer than expected?

There is inconsistency in the profiles of replication fork speed along the S phase progression in single pulse and double pulse experiments, especially in Mid and late S phase. While in the double pulse experiments replication fork speed is continuously increasing along the S phase, in single pulse experiments the speed remains constant from the 20% of the S phase to the end (Extended Figure 7B vs 7C or Figure 3D vs Extended Figure 7A). The authors should comment about these differences. Could it be related with the overestimation of replication fork speed in the double pulse experiments linked to the encounter of different forks? These events should be more frequent in Mid and late S phase.

Extended Figure 6E: The authors mention that that regions with higher read coverage have higher fork speeds and they correct this effect mathematically. However, transcribed regions replicated in early S are characterized by containing more accessible chromatin and thus, it is more likely to have higher coverage. I wonder how much of the delay observed in replication fork speed in early S is due to this correction. The authors should check the coverage distribution along the S phase progression to control for this bias. Alternatively, validation of the delay in replication fork speed in early vs late S phase could be performed by non-sequencing technologies such as DNA fibers.

The authors indicate that a recent study employing long read sequencing describe a similar early S delay in replication fork speed in human cells. Although the authors indicate that scEdU-seq has the advantage of being able to determine heterogeneity in replication dynamics between individual cells, the majority of the results presented here have been done averaging the speeds from thousands of cells. Thus, as a reader, I would like to see in the discussion, whether scEdU-seq offers additional advantages compared to similar long-read sequencing technologies, for example: cost, increase coverage, base pair resolution, simultaneous determination of other aspect of replication dynamics such as changes in replication timing...

Apart from transcriptional activity, replication fork speed could be also affected by chromatin structure and organization. The authors can perform some additional analysis of their data to explore this possibility. For example:

Extended Figure 8E indicates that in early S, replication fork speed in non-transcribed regions is faster than in transcribed regions. Is replication speed in non-transcribed regions in mid or late S as rapid as in early S? It should be the case if transcription is the only factor affecting speed but it should not if other features, for example chromatin structure or DNA compaction also affect. In the presence of DRB, is replication fork speed in early S phase equal to non-transcribed regions in non-treated cells or in non-transcribed regions of treated cells?

A main advantage of scEdU-seq compared to microscopy-based techniques is the ability to study replication fork speed in the context of the genome. Although the authors have made use of this characteristic to study replication fork speed in relation to transcription, the fork speed is calculated measuring areas of the genome that can span several kilobases. This depends on the length of the EdU pulses and the intervals between them. The authors should comment the limitation of the technique to determining replication fork speed in small regions of the genome or specific genome features like for example at promoters or enhancers.

Does the technology have enough resolution to check, for example, replication fork speed at regions of the genome containing R-loops that in principle should also reduce replication fork speed? The authors have shown that transcriptional activity reduces replication fork speed. It would be interesting to correlate this decrease with replication fork directionality. Does it correlate with transcription and replication moving in head-on orientation or co-directional orientation?

Minor comments:

Figure 2C is missing a label in the x-axis.

Author Rebuttal to Initial comments

Reviewer #1:

Remarks to the Author:

van den Berg et al. present a clever and powerful approach for analyzing replication kinetic using metabolic labeling and deep sequencing. By combining standard approaches for pulse-labeling and isolating newly replicated DNA with EdU, single-cell characterization and isolation by FACS, and genome amplification, library creation and amplification, they design a pipeline that identifies the location of replication forks in single cells. The particular power of this approach is that it allows fork rate to be inferred from the distances between signals recorded during two sequential EdU pulses. This approach to measuring fork rate is a significant advance in the field because it allows the assignment of speeds to forks at known locations on a genome-wide scale. The method is used to investigate the affect of transcription of replication fork kinetics and will be useful for many other studies in the future. I have no substantive concerns.

Thank you for the positive assessment of our work and for emphasising that scEdU-seq is a useful technology for future studies in the field.

Reviewer #2:

Remarks to the Author:

This manuscript describes a new method for measuring DNA replication and replication fork speed in single cells. The authors introduce both a single and double EdU pulse approach for this purpose. They investigate variability in replication speed along S phase and link it to transcription and DNA damage. This

is a very useful and powerful technique, the paper reports novel and interesting results, it is well written and the analyses and presentation are of very high quality.

We would like to thank the reviewer for the critical reading and their positive assessment of our manuscript. In the revised manuscript we have attempted to clarify and expand our analyses on DNA replication timing, added a comparison to scRepli-seq and included an analysis on the variability of DNA replication fork speed in single cells. We have highlighted all our changes throughout the manuscript in bold and/or magenta

I have a few questions/suggestions.

Figure 1b. Unclear why the authors chose to separate the data to early and late replication. In Repli-seq, early and late S phase fractions are collected however they are analyzed by comparing one to another rather than individually.

Thank you for the suggestion. We now use the fold change enrichment of early over late S-phase DNA replication profiles. We have added these additional analyses to our extended data figures (Fig. R2.1).

Fig. R2.1 log₂ fold change analysis of DNA Replication Timing

- a. DNA Replication Timing (50kb, log₂ fold change) for RPE-1 hTERT cells for scEdU-seq (blue, top panel) and Repli-seq (red, bottom panel) for 10 megabase on chromosome 1.
- b. Scatter plot displaying genome-wide log₂ fold change for DNA replication timing for scEdU-seq (x-axis) and Repli-seq (y-axis)

In Extended data figure 1b, the authors show lower correlation with Repli-seq for late S phase. Can they comment on that?

We have attempted to reproduce the sorting gates described by Ryba *et al.* shown in Figure 2 of their 2011 *Nature Protocols* paper. However, these gates are for another sample generated by the Gilbert lab and not for RPE-1 cells (sequencing data available on the 4D Nucleome Data Portal; <https://data.4dnucleome.org>; 4DNEXCDJJVUR). Differences between our gate settings and setting used by the Gilbert lab most likely explain the lower correlation observed for late S phase.

Extended data figure 1e and legend- this is not entirely clear. Can the authors elaborate on what the left and right plots are showing?

We have added additional explanations to the Extended Data Figure legends.

More generally, can the authors comment about how the choice of sorting gates influences their results?

The set-up of our sort gates, as shown in Extended Data Fig 1e, was a compromise between collecting enough early S-phase cells without including many non-proliferating cells in G1. We were inspired by the original sorting gates from bulk Repli-Seq experiments. However, we did sort all cells in G2 as they represent a minor population compared to G1 cells. We observe that G1 and G2 (based on DAPI intensities) contain less scEdU-seq+ cells. Especially at the lower and upper bound of these intensities we observe very low numbers of scEdU-seq+ cells (Fig. R2.2).

Fig R2.2 - DAPI intensities of QC passed cells

- EdU positive (blue) and negative cells (grey) cells following filtering on reads per bin and Coefficient of variation. The dashed lines indicate filtering thresholds.
- Histogram of DAPI intensities for both scEdU-seq positive (blue) and negative (grey) cells. The red dashed line indicates the lower threshold of DAPI intensity used for cell sorting.

Based on Figure 1f (and other figures), the data quality appears excellent, and not just for replication speed but for replication timing in general. The paper may benefit from emphasizing this aspect of the technique as well, including a quantitative comparison to other scRepli-seq approaches.

We would like to point out that we feel that scRepli-seq and scEdU-seq are complementary methods for studying DNA replication in single cells. We feel that scRepli-seq has a couple of advantages over scEdU-seq. First, scRepli-seq can be performed on fixed materials as it does not require incorporation of a synthetic uridine analog. Second, scRepli-seq can be performed at higher throughput using droplet-based methods, easily profiling 10-fold more cells compared to scEdU-seq. Finally, the availability of commercial kits for single cell DNA sequencing enables increased accessibility of scRepli-seq for the community (e.g., see recent work by Massey, D.J. et

al., Nature Comm., 2022 and Gnan, S. et al., Nature Comm., 2022. Conversely, we feel scEdU-seq has advantages over scRepli-seq. First, there is no need for copy number correction in scEdU-seq, which allows the application to more copy number heterogeneous samples. Furthermore, reagents costs of scEdU-seq, as compared to scRepli-Seq, are lower as a result of smaller reaction volumes, the lack of reliance on commercial kits and less sequencing reads required per cell resulting in 50-fold lower costs (Extended Data Table 1). Finally, scEdU-seq expands the analysis of DNA replication in single cells by quantifying DNA replication fork number and replication speeds and its variability in single cells.

There is a major methodical difference between scRepli-seq and scEdU-seq. scRepli-seq relies on whole-genome amplification, whereas scEdU-seq affinity purifies only genomic DNA containing the EdU base. This difference makes a direct comparison between these two methods challenging. We attempted to showcase the strengths of scEdU-seq to detect DNA replication forks in single cells in Fig. R2.3. First, we selected similarly staged cells from mid S-phase for scEdU-seq (Fig. R2.3a, based on S-phase Progression) to match and compare to the published scRepli-seq RPE-1 cells (N=14). Subsequently, we use a Hidden Markov model to call DNA replication forks for scRepli-seq as well as scEdU-seq. We detect approximately twice as many DNA replication forks genome-wide in scEdU-seq compared to scRepli-seq (Fig. R2.3b). Using a representative local analysis, we find that certain regions contain DNA replication forks in scEdU-seq, which are not or less frequently detected in scRepli-seq (22 Mb and 24 Mb, Fig. R2.3c,d). Thus, scEdU-seq appears to be more sensitive in detecting DNA replication forks in single cells.

Fig R2.3 Comparison of the number of DNA replication forks from scRepli-seq and scEdU-seq.

- a. Mid S-phase cells from scEdU-seq single pulse experiment for scRepli-seq comparison. DNA content (DAPI) intensities for Mid S-phase cells (blue) superimposed over cell cycle distribution of cycling RPE-1 cells (grey).
- b. Genome-wide DNA replication forks calls of single cells assayed by scRepli-seq and scEdU-seq
- c. Heatmap of scEdUseq (15 min. single EdU pulse) maximum normalized log counts for middle S-phase cells binned per 5 kb (n = 135, bottom) and summed profile (top). Heatmap of scRepli-Seq log₂ median counts of middle S-phase cells binned per 40 kb (n = 14, bottom) and summed profile (top).
- d. Quantification of the percentage of cells containing forks in indicated highlighted areas (chr2:22Mb & chr2:24Mb) for scEdU-seq (green) and scRepli-seq (blue).

That being said, based on figure 1f there does seem to be some loss of resolution at the very beginning and very end of S phase; this is expected, but can the authors comment on that?

Indeed, the quality filter in our analysis pipeline removes cells that have a low number of unique reads per cell. This could potentially remove cells with a low number of DNA replication forks, especially in early S-phase. However, to partially mitigate this, we select not only on read depth but also on the coefficient of variation to select for cells with higher signal to noise. Nonetheless, we agree that cells, which contain low amounts of incorporated EdU will be excluded from downstream analyses. Therefore, performing analysis on very early or very late S-phase cells will be challenging using only 15 minutes of EdU labelling. Note also that some cells entered or left S phase during our pulse and were therefore labelled for less than 15 minutes. We have expanded on this concept in our discussion.

In the last paragraph of the first results section, the authors talk about resolution of the technique. However, the results they present there pertain to downsampling of the data. I'm not sure if it's appropriate to refer to this analysis as "resolution". Perhaps the authors can be more specific in defining what they are analyzing here Overall, It seems to me that the authors developed a very powerful technique even just for measuring replication timing in single cells, but do not fully seize on the opportunity to showcase its power.

We agree that the term of resolution is not a completely accurate description of the results presented in Fig. 1g-h as well as Extended Data Fig. 1f. We changed the text in the Results section of our manuscript to reflect this.

The double pulse approach is clever and novel, and the correlation of replication speed with transcription levels is very interesting. It's important to note that there have been previous studies that provided evidence for accelerated replication along S phase, for instance several studies described this in the context of the inactive X chromosome. While the current study certainly improves on those earlier studies with greater resolution and at the single-cell level, it's interesting to note that it validates and generalizes those earlier predictions. In this regard, did the authors see any unusual patterns of replication speeds on the X chromosome? Analyzing the active and inactive X chromosome separately could be an interesting avenue for a follow-up study.

Thank you very much for mentioning the previous studies on the inactive X chromosome. We now also discuss this work (Koren and McCarroll, *Genome Research*, 2014) in the discussion of our manuscript and agree this is an interesting avenue for the future.

The experiments connecting replication speeds to DNA damage are very interesting. One important question is whether transcription by itself slows replication or only in the context of DNA damage. It's unclear whether the authors have answered that question fully. They suggest that DNA damage is the mechanism causing slower replication at transcribed regions, however DNA damage occurs stochastically within transcribed regions, and some forms of DNA damage (and/or repair) are more common in late compared to early S phase or in heterochromatin compared to euchromatin. It becomes a rather complex system and the author's conclusions may be somewhat of an oversimplification at the moment. One way to further understand this would be to study transcription under the DNA damage conditions the authors used (including in single cells). An additional analysis could be of the stochasticity of replication speed influences by DNA damage; the authors mention variability in the last results paragraph but it seems this section could be made clearer and/or elaborated upon.

Following the referee's suggestion, we expanded on our initial analysis of DNA replication fork speed variability in single cells. We find that inhibiting transcription has the largest overall effect on diminishing variability of DNA replication speeds within a single cell (Fig. R2.4a). In line with this, we observe that variability of DNA replication speeds is higher in early S-phase (WT DMSO), which is dramatically decreased upon the addition of the inhibitor of RNA polymerase II DRB (Fig R2.4b).

Moreover, elevated DNA damage (XRCC1Δ DMSO) increases variability of DNA replication speeds compared to steady-state (WT DMSO). Both of which are decreased following the addition of the PARP inhibitor Olaparib (Fig. R2.4a). In wild-type cells, this decrease seems to be

concentrated in early S-phase (Fig. R2.4b). Conversely, XRCC1Δ displays higher levels of variability throughout S-phase, which are diminished at all stages of S-phase with PARPi (Fig. R2.4b). This difference indicates that not only transcription but also other processes impact DNA replication speeds outside RNA polymerase II transcription. We have added these analyses to Extended Data Figure 10.

Therefore we can not exclude the impact of other chromatin bound processes (e.g. DNA damage) on DNA replication speeds and their variability within single cells. Based on these analyses and the suggestion of the referee, we have expanded the discussion in which we describe additional hypotheses, which might explain the acceleration of DNA replication during S-phase.

Fig R2.4 Variability of DNA replication speeds in single cells and over S-phase

- a. Variability of DNA replication speeds within single cells (kb/min, y-axis) split over Wild-type and XRCC1Δ in RPE-1 cells labelled with $\Delta t = 75$ min. EdU scheme.
- b. Variability of DNA replication speeds within single cells (kb/min, y-axis) over S-phase Progression (x-axis) split over wild-type DMSO, wild-type DRB, wild-type PARPi, XRCC1Δ DMSO and XRCC1Δ PARPi conditions in RPE-1 cells labelled with $\Delta t = 75$ min. EdU scheme.

A limitation of the study is its being confined to a single cell line (RPE-1).

To expand our observations beyond RPE-1 cells, we have added a new experimental data set using human induced pluripotent stem cells (hiPSC, Fig. R2.5a). Similarly, we find that these cells show comparable speeds to RPE-1 hTERT cells (Fig. R2.5b) Moreover, these cells also show an increase in DNA replication speeds over S-phase (Fig. R2.5c). In addition, hiPSCs also display similar replication fork variability within single cells (Fig. R2.5d)

Fig. R2.5 DNA replication speeds, acceleration and variability of speeds in hiPSC

- Heatmap of scEdU-seq maximum normalized log counts for $\Delta t=75\text{min}$ human induced pluripotent stem cells (hiPSC) ordered according to S-phase progression (y-axis) and binned per 5kb bins (x-axis) for 20 Megabase of chromosome 2
- Violin plot depicting DNA replication speeds in hiPSC (every dot indicates a single cell)

- c. DNA replication speed over S-phase in hiPSCs treated with DMSO subjected to $\Delta t = 75$ min. labeling scheme. Speed estimates in kb/min (y-axis) over S-phase progression (x-axis). Every dot is a cell, the line indicates a rolling-window median smooth and the ribbon the standard deviation around the median
- d. Variability of DNA replication speeds within single hiPSC cells (kb/min, y-axis) in hiPSC cells labelled with $\Delta t = 75$ min. EdU scheme.

Reviewer #3:

Remarks to the Author:

van den Berg et al. have developed the scEdU-seq technology that, for the first time, allows to study replication fork progression in the context of the genome and interrogates about the cell-to-cell variability of this process in human cells. This technology is easily accessible to any laboratory and represents a significant advancement in the fields of DNA replication and repair as well as being relevant to any other field studying biological processes linked to cell division. The results presented in this study are intriguing and demonstrate how replication fork speed is increasing during the S phase and showed that the delay in early S can be, in part, due to the repair of DNA damage associated to the transcriptional process. However, in this manuscript, I would appreciate a better comparison with similar long-read sequencing technologies. Moreover, considering that the primary advantage of the technique is to obtain genomic information and cell to cell variability of the replication fork progression, a more detailed analysis of the resolution to determine replication fork speed at specific loci or genome features should be addressed. Finally, the authors should provide additional methodology details in the results section to facilitate the reading.

We would like to thank the reviewer for this positive assessment and for critically reading our manuscript. In the revised manuscript we have now incorporated a comparison to long-read sequencing, included an analysis on the resolution of scEdU-seq and provided more details in the Methods section of the manuscript. We have highlighted all our changes throughout the manuscript in bold and/or magenta

Main comments:

Figure 1A. When describing the methodology, the authors should indicate that they used cell sorting before performing single cell reactions. As it is written is not clear for the reader.

We have now included an additional explanation in the Result section of the main text to explain the steps preceding the single cell reactions.

Results from Figure 2E and F, showed that replication fork speed values are higher when the time between pulses is longer. I wonder whether the replication fork speed can be overestimated when the time between pulses is long because the longer the pulses are separated, the higher is the chance that two forks converge and are consider as one single fork instead of two. Could this bias be corrected by filtering out forks in which the width of any of the two EdU pulses is longer than expected?

This is indeed correct and excluding interfering signals of converging forks by filtering out wider DNA replication forks is an interesting idea. We explored this strategy but unfortunately, we do not observe a clear bimodal distribution in the $\Delta t = 75$ min. double pulse experiment when compared to the single pulse experiment (Fig. R3.1, bottom panel). As expected, we do observe that the width of a double pulse experiment increases compared to single pulse (Fig. R3.1, top panel), however it is very difficult to systematically identify a threshold that separates single forks from converged forks. We now discuss this phenomenon in the discussion section of the manuscript.

Fig. R3.1 DNA replication fork widths in single and double pulse experiments.

Top panel. Boxplot of DNA replication fork widths (\log_{10}) from $\Delta t = 75$ min. (yellow) and single pulse (grey) Bottom panel - Maximum normalised histogram of DNA replication fork widths (\log_{10})

There is inconsistency in the profiles of replication fork speed along the S phase progression in single pulse and double pulse experiments, especially in Mid and late S phase. While in the double pulse experiments replication fork speed is continuously increasing along the S phase, in single pulse experiments the speed remains constant from the 20% of the S phase to the end (Extended Figure 7B vs 7C or Figure 3D vs Extended Figure 7A). The authors should comment about these differences. Could it be related with the overestimation of replication fork speed in the double pulse experiments linked to the encounter of different forks? These events should be more frequent in Mid and late S phase.

We hypothesise that this is related to the manner in which we quantify single versus double-pulse experiments. First, in double-pulse experiments replication speeds are measured over a longer period compared to single-pulse experiments. In a double-pulse experiment the replication speed represents an average over 75 minutes, whereas in a single-pulse experiment the replication speed is averaged over 15 minutes. Second, the analysis of single-pulse data is based on the median width of all DNA replication forks in a single cell, which is dependent on

the sequencing depth of each experiment (as highlighted in Extended Data Fig. 3). Moreover, the accuracy of the double-pulse strategy is higher than the single-pulse strategy and the single-pulse systematically under-estimates replication speed (Extended Data Fig. 3). It is therefore challenging to make quantitative statements on the difference between the single and double-pulse experiments. However, it is encouraging that both strategies show the same trend: acceleration of replication.

Extended Figure 6E: The authors mention that regions with higher read coverage have higher fork speeds and they correct this effect mathematically. However, transcribed regions replicated in early S are characterised by containing more accessible chromatin and thus, it is more likely to have higher coverage.

We would like to clarify that our use of the word coverage does not relate to the classical meaning of sequencing reads per genomic region. We meant to say that the part of the genome which is covered by a DNA replication fork (with a certain width). To avoid future confusion, we have changed the wording from “coverage” to “longer DNA replication tracks”. We apologise for this unfortunate choice of wording.

Of note, in contrast to chromatin-based methods (e.g. ChIP-seq), which use sonication or MNase treatment without protein digestion, we employ a proteinase K step prior to restriction enzyme digestion of the genome. We observe less variability in scEdU-seq-based input coverage between 5 kb bins compared to ChIP-seq input (Fig. R3.2a). This results in more equal coverage of the genome and not a bias to certain types of chromatin (Fig. R3.2b).

Fig. R3.2 Coverage per pulse versus DNA replication fork width

- Histogram depicting the number of reads (x-axis, i.e., coverage) per 5kb bin from scEdU-seq input (blue) and ChIP input (grey) for RPE-1 hTERT cells
- Bargraph depicting the number of reads (x-axis, i.e., coverage) per 5kb bin from scEdU-seq input (bottom, blue) and ChIP input (top, grey) over 40 Mb of Chromosome 2 for RPE-1 hTERT cells.

I wonder how much of the delay observed in replication fork speed in early S is due to this correction. The authors should check the coverage distribution along the S phase progression to control for this bias.

In order to address this question, we analysed the coverage (reads per 5kb bin) along S-phase Progression quantiles using a scEdU-seq input sample. We observe that read coverage per 5 kilobase does not change along S-phase Progression (Fig. R3.3). Based on this analysis, we strongly believe that the coverage does not change across S-phase using scEdU-seq library preparation methods. More importantly, we feel that the coverage along S-phase does not impact speed differences, otherwise, we would have difficulty detecting higher DNA replication speeds in DRB and PARP inhibitor treated cells. In addition, employing Nanopore sequencing by the Boemo lab found lower DNA replication speeds in early S-phase. Finally, we performed DNA fiber analysis and found longer IdU track lengths for RPE-1 hTERT cells from late S-phase (Fig. R3.4 and see point below)

Fig. R3.3 Coverage of 5kb bins along S-phase Progression

scEdU-seq input samples were binned at a 5kb size and reads inside bins were counted. Subsequently, the bins were split by the S-phase progression of said bin and number of reads were plotted for each S-phase progression quantile

Alternatively, validation of the delay in replication fork speed in early vs late S phase could be performed by non-sequencing technologies such as DNA fibers.

In order to address this question, we performed a cell sort using RPE-1 hTERT FUCCI cells, which we label for 20 minutes with 250mM IdU. We sort two population, early and late S-phase cells (yellow and green for early and late respectively, Fig. R3.4a), and perform DNA combing and fiber analyses on these samples. We find that IdU fiber lengths in early S-phase are shorter when compared to late S-phase cells (Fig. R3.4b). We have added these data to Figure 3 of our manuscript.

Fig. 3.4 IdU track length analysis by DNA combing and fiber analysis

a. Early (yellow) and Late (green) S-phase sorted RPE-1 hTERT FUCCI cells superimposed over a 2D density distributed of all detected single cell events during fluorescence activated cell sorting (FACS) b. Detected DNA fiber length analysis by DNA combing analysis from indicated cell population labeled with IdU (20min. labeling).

The authors indicate that a recent study employing long read sequencing describe a similar early S delay in replication fork speed in human cells. Although the authors indicate that scEdU-seq has the advantage of being able to determine heterogeneity in replication dynamics between individual cells, the majority of the results presented here have been done averaging the speeds from thousands of cells. Thus, as a reader, I would like to see in the discussion, whether scEdU-seq offers additional advantages compared to similar long-read sequencing technologies, for example: cost, increase coverage, base pair resolution, simultaneous determination of other aspect of replication dynamics such as changes in replication timing...

We have added a discussion on the use of scEdU-seq compared to Nanopore sequencing to the Discussion section.

Apart from transcriptional activity, replication fork speed could be also affected by chromatin structure and organisation. The authors can perform some additional analysis of their data to explore this possibility. For example:

Extended Figure 8E indicates that in early S, replication fork speed in non-transcribed regions is faster than in transcribed regions. Is replication speed in non-transcribed regions in mid or late S as rapid as in early S?

In order to visualise effects of other processes on DNA replication speeds outside of transcription, we filtered out DNA replication speeds originating from transcribed regions in the genome. Subsequently, we divided cells into four groups along S-phase Progression and plotted their respective DNA replication speeds. We observe that DNA replication speeds along the progression throughout S-phase, even in the absence of transcription, still increases (Fig. R3.5). This is in line with our initial observation that DNA replication speeds along S-phase continually accelerate. However, it does suggest there are, possibly, alternative processes which impact DNA replication speeds throughout S-phase.

Fig. R3.5 DNA replication speeds outside transcribed regions along S-phase

Maximum-normalized counts (y-axis) of DNA replication speeds (x-axis) outside of transcribed regions along S-phase (quantiles are indicated by the colors)

It should be the case if transcription is the only factor affecting speed but it should not if other features, for example chromatin structure or DNA compaction also affect.

To expand on the previous analysis along S-phase progression, we set out to quantify DNA replication speeds in regions with different types of chromatin modifications. In order to perform these analyses, we profiled H3K9me3, K3K27me3 and H3K36me3 in bulk on RPE-1 hTERT. Next, we quantified DNA replication speeds inside these regions with different chromatin modifications (Fig. R3.6a - orange lines), akin to the analysis presented in Extended Data Fig. 8e. As expected, we observe that H3K36me3 has a similar effect on DNA replication speeds as transcribed regions confirming our initial observation. Conversely, we find that H3K9me3-repressed chromatin confers higher DNA replication speeds. In addition, we observe that inside H3K27me3 chromatin (orange line) DNA

replication speeds do not drastically differ from outside H3K27me3 regions (Fig. R3.6a - grey line). In addition, we quantify the difference of DNA replication speeds (Δ RFS) by subtracting the DNA replication speeds from outside (grey) the target region from target region (orange) DNA replication speeds. Finally, we confirm the distribution of these histone marks along S-phase (Fig. R3.6b). We observe that H3K36me3 chromatin mainly occurs in early replicating regions, whereas H3K9me3 occupies late replicating regions. In addition, we find that H3K27me3 chromatin replicates throughout S-phase but peaks in mid-S-phase. We thank the reviewer for their insight and feel that these observation are an interesting avenue for a follow-up study

Fig. R3.6 DNA replication speeds in H3K36me3, H3K9me3 and H3K27me3 chromatin

- Maximum-normalized counts (y-axis) of DNA replication speeds (x-axis) outside of transcribed regions along S-phase (quantiles are indicated by the colors). Adjusted p-values with pairwise T-test are; H3K27me3 (0.512, ns), H3K36me3 (3.0e-03, ***), H3K9me3 (3.9e-4, ****) and Transcribed (1.2e-04, ****)
- Relative chromatin mark abundance (y-axis) along S-phase progression (x-axis) split by H3K36me3, H3K9me3 and H3K27me3

In the presence of DRB, is replication fork speed in early S phase equal to non-transcribed regions in non-treated cells or in non-transcribed regions of treated cells?

Thank you for the suggestion for this analysis. In order to address this question, we perform a similar analysis as Extended Data Fig. 8e. We analyse DNA replication speeds in DMSO or DRB treated RPE-1 cells for transcribed regions as well as H3K36me3 chromatin (Fig. R3.7). We find that we can increase DNA replication speeds, in transcribed as well as H3K36me3 chromatin, using

the RNAP2 inhibitor DRB. We find that the RNAP2 inhibitor almost completely removes the influence of transcription or H3K36me3 on DNA replication speeds.

Fig. R3.7 H3K36me3, H3K9me3 and H3K27me3 chromatin mark distribution along S-phase

Maximum-normalized counts (y-axis) of DNA replication speeds (x-axis) outside of transcribed regions along S-phase (quantiles are indicated by the colors) for DMSO (left) or DRB treated RPE-1 cells. Adjusted p-values with pairwise T-test are; DMSO H3K36me3 (3.0e-03, ***), DMSO Transcribed (1.2e-04, ****), DRB H3K36me3 (1.1e-02, **), DRB Transcribed (0.398, *ns*)

A main advantage of scEdU-seq compared to microscopy-based techniques is the ability to study replication fork speed in the context of the genome. Although the authors have made use of this characteristic to study replication fork speed in relation to transcription, the fork speed is calculated measuring areas of the genome that can span several kilobases. This depends on the length of the EdU pulses and the intervals between them. The authors should comment on the limitation of the technique to determine replication fork speed in small regions of the genome or specific genome features like for example at promoters or enhancers.

Indeed, scEdU-seq cannot quantify replication speed in small regions of the genome. During 15 minutes of EdU pulse, we label approximately a region of 30 kb which is much larger than the size of typical promoters and enhancers (approximately 0.8 to 1 kb in length, *Arensbergen, J., et al., 2016 Nature Biotechnology, 2016*). We have expanded on the limitations of scEdU-

seq to detect DNA replication speeds over small functional units in the genome in our discussion. In addition, we have highlighted the use of Nanopore sequencing in detecting replication speeds as well as stalling events over small regions such as promoters and enhancers.

Does the technology have enough resolution to check, for example, replication fork speed at regions of the genome containing R-loops that in principle should also reduce replication fork speed?

Similar to the size of enhancer and promoters, several reports state that R-loops are between 0.2 to 2.0 kb in length (*Crossley, M. P., Bocek, M. & Cimprich, K. A. Mol. Cell, 2019*). Following a similar argument in the previous question, we feel that scEdU-seq is not the preferred method to detect these events. Nanopore-based methods are better suited for this, however this technology is not yet possible with single-cell resolution. We feel that scEdU-seq and Nanopore based technology are highly complementary. As mentioned before, we now highlight this complementarity in the discussion of the manuscript.

The authors have shown that transcriptional activity reduces replication fork speed. It would be interesting to correlate this decrease with replication fork directionality. Does it correlate with transcription and replication moving in head-on orientation or co-directional orientation?

We agree with the reviewer that this is an extremely interesting avenue for further research. However, given the current sensitivity of detecting DNA replication speeds in specific genomic features (see discussion above), we feel that this question is better addressed using inhibitor or perturbation experiments which directly impact 'co-directional' or 'head-on' R-loops. Using for instance, a rapid degradation system for components of the integrator complex or RNase H1. Ideally, these analyses would also include the detection of nascent RNA from the same single cells. Taken together, we feel that these experiments are fantastic opportunities for future avenues.

Minor comments:

Figure 2C is missing a label in the x-axis.

Thank you for noticing this formatting error, we have added a label for the x-axis of Fig. 2C.

Decision Letter, first revision:

Our ref: NMETH-A54035A

10th Apr 2024

Dear Dr. van Oudenaarden,

Thank you for submitting your revised manuscript "Quantifying DNA replication speeds in single cells by scEdU-seq" (NMETH-A54035A). It has now been seen by the original referees and their comments are below. Since reviewer#2 is not available anymore, we have consulted with reviewer#1 about your responses to reviewer#2. The reviewers find that the paper has improved in revision, and therefore we'll be happy in principle to publish it in Nature Methods, pending minor revisions to comply with our editorial and formatting guidelines.

TRANSPARENT PEER REVIEW

Please note: we allow redactions to authors' rebuttal and reviewer comments in the interest of confidentiality. If you are concerned about the release of confidential data, please let us know specifically what information you would like to have removed. Please note that we cannot incorporate redactions for any other reasons. Reviewer names will be published in the peer review files if the reviewer signed the comments to authors, or if reviewers explicitly agree to release their name. For more information, please refer to our FAQ page.

ORCID

Sincerely,
Lei

Lei Tang, Ph.D.
Senior Editor
Nature Methods

Reviewer #3 (Remarks to the Author):

I appreciate the efforts of the authors to address all my concerns. This is a very nice piece of work describing a powerful technique and I believe it deserves to be published in this journal.

Author Rebuttal, first revision:

[There is no rebuttal letter at this stage.]

Final Decision Letter:

17th May 2024

Dear Dr van Oudenaarden,

I am pleased to inform you that your Article, "Quantifying DNA replication speeds in single cells by scEdU-seq", has now been accepted for publication in Nature Methods. The received and accepted dates will be 4th Oct 2023 and 17th May 2024. This note is intended to let you know what to expect from us over the next month or so, and to let you know where to address any further questions.

Over the next few weeks, your paper will be copyedited to ensure that it conforms to Nature Methods style. Once your paper is typeset, you will receive an email with a link to choose the appropriate publishing options for your paper and our Author Services team will be in touch regarding any additional information that may be required. It is extremely important that you let us know now whether you will be difficult to contact over the next month. If this is the case, we ask that you send

us the contact information (email, phone and fax) of someone who will be able to check the proofs and deal with any last-minute problems.

Please note that *Nature Methods* is a Transformative Journal (TJ). Authors may publish their research with us through the traditional subscription access route or make their paper immediately open access through payment of an article-processing charge (APC). Authors will not be required to make a final decision about access to their article until it has been accepted. Find out more about Transformative Journals

If you are active on Twitter/X, please e-mail me your and your coauthors' handles so that we may tag you when the paper is published.

You can now use a single sign-on for all your accounts, view the status of all your manuscript submissions and reviews, access usage statistics for your published articles and download a record of

your refereeing activity for the Nature journals.

Best regards,
Lei

Lei Tang, Ph.D.
Senior Editor
Nature Methods